# Cognitive Mirrors: Exploring the Diverse Functional Roles of Attention Heads in LLM Reasoning

**Xueqi Ma**[1]  **Jun Wang**[1,3]*  **Yanbei Jiang**[1]  **Sarah Monazam Erfani**[1]
**Tongliang Liu**[2]  **James Bailey**[1]
[1]The University of Melbourne  [2]The University of Sydney
[3]Amazon
{xueqim, jun2, yanbeij}@student.unimelb.edu.au
{sarah.erfani, baileyj}@unimelb.edu.au
tongliang.liu@sydney.edu.au

## Abstract

Large language models (LLMs) have achieved state-of-the-art performance in a variety of tasks, but remain largely opaque in terms of their internal mechanisms. Understanding these mechanisms is crucial to improve their reasoning abilities. Drawing inspiration from the interplay between neural processes and human cognition, we propose a novel interpretability framework to systematically analyze the roles and behaviors of attention heads, which are key components of LLMs. We introduce CogQA, a dataset that decomposes complex questions into step-by-step subquestions with a chain-of-thought design, each associated with specific cognitive functions such as retrieval or logical reasoning. By applying a multi-class probing method, we identify the attention heads responsible for these functions. Our analysis across multiple LLM families reveals that attention heads exhibit functional specialization, characterized as cognitive heads. These cognitive heads exhibit several key properties: they are universally sparse, and vary in number and distribution across different cognitive functions, and they display interactive and hierarchical structures. We further show that cognitive heads play a vital role in reasoning tasks—removing them leads to performance degradation, while augmenting them enhances reasoning accuracy. These insights offer a deeper understanding of LLM reasoning and suggest important implications for model design, training and fine-tuning strategies. The code is available at https://github.com/sihuo-design/CognitiveMirrors.

## 1  Introduction

Large language models (LLMs) [1, 14, 30, 38], built on neural networks that mimic the structure of the human brain, have demonstrated exceptional performance across various natural language processing (NLP) tasks, often exceeding human capabilities. This has sparked growing interest in exploring the potential similarities between the cognitive processes of LLMs and the human brain. Prior studies have demonstrated that LLMs can predict brain responses to natural language [8, 28], indicating a functional alignment between artificial models and biological systems. However, to the best of our knowledge, systematic efforts to align reasoning processes between LLMs and human cognitive agents remain scarce. When solving complex reasoning tasks (e.g., a mathematical multiple-choice question; Figure 1), the human brain engages a network of specialized regions: the frontal lobe recalls relevant knowledge [35], language areas (e.g., Wernicke's and Broca's) support semantic processing [23, 20], and the parietal and prefrontal cortices carry out higher-order reasoning [5, 15].

---

*This work is not related to Amazon.

39th Conference on Neural Information Processing Systems (NeurIPS 2025).

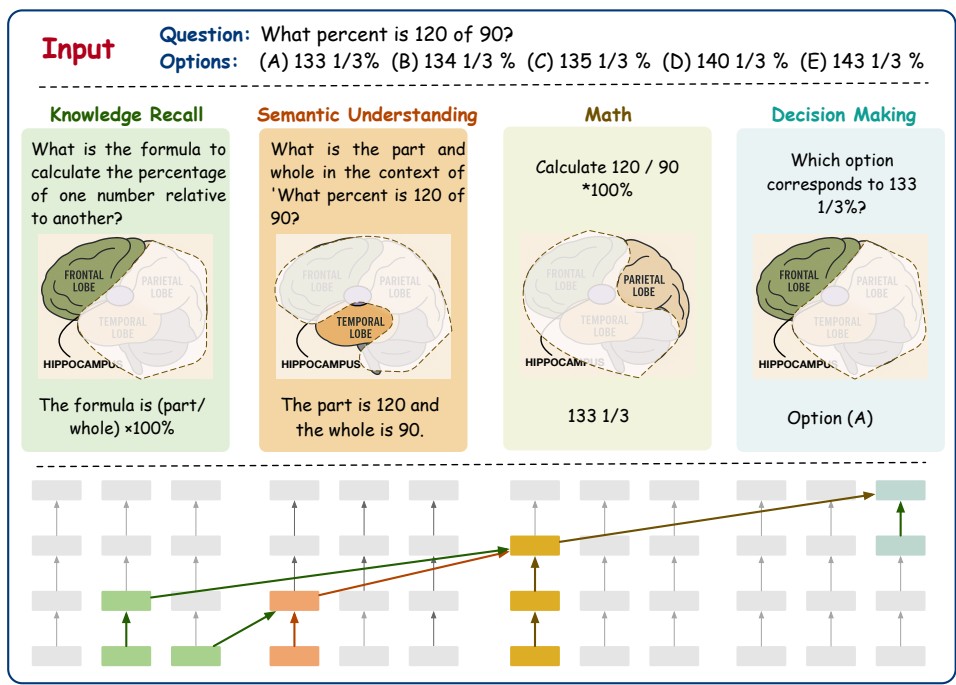

Figure 1: To solve a complex question, the human brain engages multiple regions to perform distinct cognitive functions necessary for generating a response. We explore whether there are specific attention heads in LLM play functional roles in producing answers.

Analogously, recent research suggests that components within LLMs may also take on specialized roles. For example, multi-head attention mechanisms in transformers [31] have been found to handle distinct functions, such as information retrieval [36] or maintaining answer consistency [17], pointing toward a form of architectural division of labor. However, most of these findings are based on relatively simple tasks [40], leaving open how such specialization operates under complex, multi-step reasoning scenarios. In parallel, prompting techniques like chain-of-thought (CoT) [34] have been shown to improve LLM performance by decomposing complex problems into intermediate steps, a strategy reminiscent of human problem-solving, like the example in Figure 1. We hypothesize that such prompting may activate and coordinate specialized components within the model. Thus, analyzing the behavior of attention heads under CoT reasoning could contribute insights for a deeper understanding of the internal workings of LLMs and how they process complex tasks.

In this work, we present a novel interpretability framework to systematically analyze the cognitive roles of attention heads during complex reasoning. To facilitate this, we introduce Cognitive Question&Answering (CogQA), a benchmark dataset that decomposes natural language questions into structured subquestions annotated with fine-grained cognitive functions, such as retrieval, logical inference, and knowledge recall. Leveraging CogQA, we develop a multi-class probing method to identify and characterize attention heads responsible for distinct cognitive operations within the transformer architecture.

We conduct extensive experiments on three major LLM families, including LLaMA [30], Qwen [38], and Yi [39]. Our results reveal the existence of cognitive heads that consistently exhibit **universality**, **sparsity**, and **layered functional organization** across architectures. Further analysis of the correlations among these cognitive heads reveals clear **functional clustering**, with heads grouping based on cognitive roles, and uncovers a **hierarchical structure** in which lower-level heads modulate higher-level ones—mirroring the modular and distributed processing observed in the human cortex [5, 23].

Furthermore, we validate the functional importance of these heads by showing that their removal degrades performance on complex tasks and leads to specific error patterns, while their enhancement improves reasoning capabilities. Our findings shed light on the structured cognitive architecture embedded in LLMs and open avenues for function-aware model design and analysis.

## 2 CogQA

In this section, we present a detailed account of our benchmark dataset CogQA's construction and key characteristics. Although extensive existing benchmark collections span a wide array of NLP tasks, to our knowledge no resource explicitly evaluates LLM reasoning across diverse cognitive functions. To address this, we introduce CogQA, a dataset containing 570 main questions and 3,402 subquestions. Each example comprises a question, its answer, and an annotation specifying the cognitive function required for resolution.

### 2.1 Cognitive Function

To systematically capture the cognitive processes involved in complex reasoning tasks, we categorize cognitive functions into two groups: low-level functions and high-order functions, inspired by established frameworks in cognitive science [4, 12]. Low-level functions primarily involve information retrieval and linguistic analysis, while high-order functions engage more abstract reasoning, problem-solving, and decision-making. Detailed descriptions of these cognitive functions are provided in Appendix A.4.

The low-level cognitive functions include:

- **Retrieval**: locating relevant information from an external source or prior context.
- **Knowledge Recall**: accessing stored factual or procedural knowledge from memory.
- **Semantic Understanding**: interpreting the meaning of words, phrases, or concepts.
- **Syntactic Understanding**: analyzing the grammatical structure of a sentence.

The high-order cognitive functions include:

- **Mathematical Calculation**: performing arithmetic or numerical operations.
- **Logical Reasoning**: drawing conclusions based on formal logical relationships.
- **Inference**: deriving implicit information that is not directly stated.
- **Decision-Making**: selecting the best outcome among alternatives based on reasoning.

This categorization reflects a natural progression from basic information processing to complex cognitive integration. Both the human brain and LLMs encompass a wide range of functional modules. Our focus in this work is specifically on reasoning-related cognitive functions. By identifying and organizing these eight core reasoning functions, we can more clearly examine how LLMs handle different types of thinking steps, in a way that is both systematic and easy to interpret.

### 2.2 Data Collections

Based on our categorization of cognitive functions, we sampled 750 diverse questions from NLP reasoning benchmarks, selecting 150 examples from each of AQuA [18], CREAK [24], ECQA [2], e-SNLI [7], and GSM8K [11]. These datasets cover a range of reasoning types, including logical, mathematical, and commonsense reasoning. Using the CoT paradigm, we prompted GPT-4o [16] to decompose each question into subquestions, each targeting a single cognitive function. The prompt encourages structured, step-by-step reasoning, with each subquestion being clear, answerable, and sequentially dependent. This yields a set of subquestion-answer-cognitive function (subQAC) triples for each QA pair: $\text{subQACs} = \{(q_i, a_i, c_i)\}_{i=1}^{k}$, where each contains a subquestion $q_i$, its concise answer $a_i$, and the corresponding cognitive function label $c_i$. The prompt for generating subquestions and examples are list in Appendix A.4 and Appendix A.6, respectively.

### 2.3 Data Filtering and Annotation

Recent advances have made it increasingly feasible to use LLMs for dataset construction, owing to their strong reasoning abilities and capacity to generate high-quality annotations at scale [33]. Although our dataset is constructed automatically using an LLM to reduce manual effort, we implement a strict two-stage human verification pipeline to ensure data quality and mitigate hallucinations. In the first stage, three expert annotators independently assess whether the subquestions are logically structured and align with natural human reasoning. QA pairs with inconsistent or incoherent decompositions are filtered out. In the second stage, annotators verify and, if necessary, relabel the cognitive function associated with each subquestion to ensure alignment with the intended mental process.

Finally, we validate the subanswers by cross-checking them using the GPT-o4-mini model [25], followed by human adjudication where discrepancies arise. Details of the annotation process and rubric can be found in Appendix A.5. This multi-step filtering ensures that each retained subQAC triple reflects a coherent, interpretable reasoning step grounded in core cognitive functions. After this refinement, our final dataset contains 570 main QA and 3,402 validated subQAC triplets.

# 3 Cognitive Function Detections

Given the CogQA dataset, we aim to identify which attention heads in LLMs are associated with specific cognitive functions. We adopt a probing-based framework, a widely used interpretability technique in which an auxiliary classifier is trained to predict properties from intermediate model representations [3, 6, 29]. We frame this as a multi-class classification task: for each cognitively annotated subquestion, we extract head activations (see Section 3.1), train classifier and compute importance scores to identify contributing heads (see Section 3.2). Unlike prior work focusing on a single-class, our method captures many-to-many relationships between heads and functions, enabling a more detailed analysis of functional specialization and overlap compared to prior single-class approaches.

## 3.1 Head Feature Extraction

Given a large language model $\mathcal{M}$, we generate an answer $a_i^{\mathcal{M}}$ for each subquestion $q_i$ derived from a main question $Q_i$. To support coherent multi-step reasoning, we include preceding subquestions and their answers as contextual input, emulating the incremental reasoning process observed in human cognition.

During inference, input tokens are embedded and processed through successive transformer layers. At each layer, attention and feedforward operations update the residual stream, which is ultimately decoded into token predictions. For each generated token $i$, we extract attention head outputs $X_i = \{x_l^m(i) \mid l = 1, \ldots, L, \ m = 1, \ldots, M\}$ across all layers, where $x_l^m$ denotes the value vector from the $m$-th head in layer $l$ projected into the residual stream, with $M$ the number of heads per layer and $L$ the total number of layers.

Let $N_t$ denote the number of tokens in the generated answer $a_i^{\mathcal{M}}$. To isolate semantically informative content relevant to reasoning, we select the top-$k$ most important tokens, [2] determined by prompting GPT-o4-mini [25] (skilled in reasoning), yielding an index set $\mathcal{I}_k$ with $|\mathcal{I}_k| = k$ (Top-$k$ ($k = 5$) token examples are in Appendix A.10). For each index $j \in \mathcal{I}_k$, we extract the corresponding attention head activations $X_j$, and compute the averaged activation feature for the $m$-th head in layer $l$ as $\bar{x}_l^m = \frac{1}{k} \sum_{j \in \mathcal{I}_k} x_l^m(j)$. This results in a full set of head-level features $\bar{X} = \{\bar{x}_l^m \mid l = 1, \ldots, L, \ m = 1, \ldots, M\}$.

Given prior findings suggesting that cognitive functions may vary by layer depth [40], we incorporate layer-wise information by computing the average activation $\bar{x}_l = \frac{1}{M} \sum_{m=1}^{M} \bar{x}_l^m$ for each layer. We then augment each head-level vector with its corresponding layer summary, resulting in enriched features $\bar{x}_l^{m'} = [\bar{x}_l^m; \bar{x}_l]$. For each subQA triplet $(q_i, a_i, c_i)$, the final input to the probing classifier is given by $\{\bar{x}_l^{m'} \mid l = 1, \ldots, L, \ m = 1, \ldots, M\}$.

## 3.2 Heads Importance

For the CogQA dataset with $N$ subQA pairs, we collect all activations to construct the probing dataset:

$$\mathcal{D}_{\text{probe}} = \left\{ (\bar{x}_l^{m'}, \ c)_i \right\}_{i=1}^{N}, l \in \{1, \ldots, L\}, \ m \in \{1, \ldots, M\} \tag{1}$$

We split the dataset into training and validation sets with a $4{:}1$ ratio. Each attention head feature is first passed through a trainable linear projection for dimensionality reduction, followed by a two-layer MLP that performs multi-class classification over cognitive functions (training details are

---

[2]We include an ablation study in Appendix A.9 to analyze the impact of using alternative token positions.

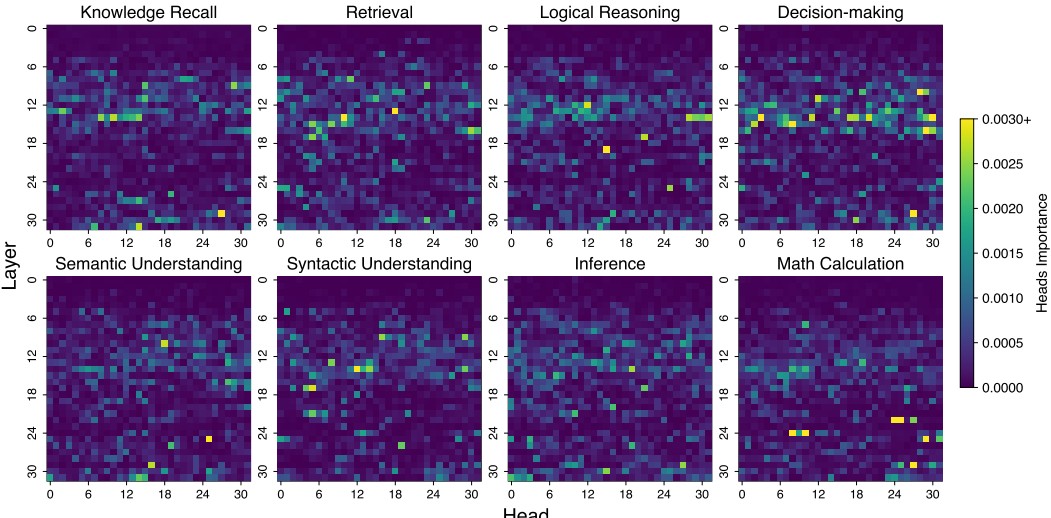

Figure 2: The existence of cognitive heads in Llama3.1-8B-instruct responsible for eight distinct functions in complex reasoning tasks. The x-axis represents the head index, while the y-axis indicates the layer index.

provided in Appendix A.3). To interpret the contribution of individual heads to each function, we use a gradient-based attribution method. Specifically, for each function class $c$, we compute the contribution of each head feature via the gradient×activation technique:

$$I_j^{(c)} = \mathbb{E}_{(\bar{x},c)\sim\mathcal{D}_{\text{probe}}} \left[ \frac{\partial \hat{y}_c}{\partial \bar{x}_j} \cdot \bar{x}_j \right], \tag{2}$$

where $\bar{x}_j$ is the $j$-th head input feature, and $\hat{y}_c$ is the classifier's predicted logit for class $c$. This yields an importance score for each attention head with respect to each cognitive function. We aggregate the scores into a matrix $\mathbf{I} \in \mathbb{R}^{C \times (L \cdot M)}$, where each row corresponds to a function class and each column to a specific head in a specific layer.

We hypothesize that attention heads with higher importance scores contribute more significantly to each cognitive function. By ranking heads according to their importance, we can identify which heads and layers are specialized for specific functions. Subsequent targeted interventions on these heads validate the effectiveness of this approach.

## 4 Experiments

We conduct a series of experiments on three LLM families across various model scales, including LLaMA [30] (Llama3.1-8B-instruct and Llama3.2-3B-instruct), Qwen [38] (Qwen3-8B and Qwen3-4B), and Yi [39] (Yi1.5-9B and Yi1.5-6B). Our goal is to identify cognitive attention heads associated with specific reasoning functions and evaluate their roles via targeted interventions. By selectively masking these heads, we assess their functional significance in supporting downstream performance. We evaluate our method in terms of functional alignment, consistency across models, and causal impact on reasoning tasks. Results confirm the existence of sparse, function-specific heads and highlight their critical contribution to structured cognitive processing within LLMs.

### 4.1 Properties of Cognitive Heads

Our analysis reveals that cognitive head importance in large language models exhibits three key properties: **sparsity and universality**, and **layered functional organization**. To illustrate these characteristics, we present the heatmap of attention head importance scores across eight cognitive functions in Llama3.1-8B-instruct (Figure 2).

Table 1: Intervention results (%) of cognitive heads vs. random heads across 8 cognitive functions: **Retrieval**, Knowledge **Recall**, **Semantic** Understanding, **Syntax** Understanding, **Math** Calculation, Inference, **Logic** Reasoning, and **Decision** Making. Lower values indicate more effective intervention outcomes, suggesting that the corresponding heads play a greater role in the cognitive function.

| Model | Inter_Head | Information Extraction and Analysis Functions | | | | | | | | Higher-Order Processing Functions | | | | | | | |
| | | Retrieval | | Recall | | Semantic | | Syntactic | | Math | | Inference | | Logic | | Decision | |
| | | comet | acc | comet | acc | comet | acc | comet | acc | comet | acc | comet | acc | comet | acc | comet | acc |
|---|---|---|---|---|---|---|---|---|---|---|---|---|---|---|---|---|---|
| Llama3.1-8B | random | 90.83 | 84.71 | 87.85 | 83.84 | 91.44 | 97.50 | 87.81 | 66.17 | 94.25 | 83.08 | 91.90 | 70.18 | 91.39 | 54.69 | 97.64 | 90.91 |
| | cognitive | 44.96 | 8.24 | 56.93 | 38.38 | 81.98 | 75.00 | 69.20 | 40.00 | 87.81 | 66.17 | 76.65 | 52.63 | 52.07 | 4.69 | 56.02 | 4.55 |
| Llama3.2-3B | random | 87.89 | 86.47 | 76.35 | 68.69 | 90.54 | 90.00 | 75.82 | 40.00 | 94.98 | 69.65 | 95.66 | 85.96 | 92.75 | 76.56 | 93.30 | 81.82 |
| | cognitive | 49.47 | 17.06 | 49.69 | 13.13 | 52.29 | 10.00 | 43.62 | 0.00 | 92.01 | 80.10 | 53.60 | 7.02 | 46.69 | 0.00 | 49.25 | 0.00 |
| Qwen3-8B | random | 92.81 | 75.29 | 89.90 | 53.54 | 92.73 | 42.50 | 88.60 | 80.00 | 92.69 | 60.20 | 94.45 | 24.56 | 94.15 | 20.31 | 96.52 | 31.82 |
| | cognitive | 59.19 | 38.24 | 64.81 | 30.30 | 85.95 | 47.50 | 46.26 | 0.00 | 89.29 | 53.23 | 72.77 | 35.09 | 87.61 | 21.88 | 83.17 | 54.55 |
| Qwen3-4B | random | 94.17 | 84.71 | 84.61 | 77.78 | 86.91 | 77.50 | 98.15 | 80.00 | 87.15 | 44.78 | 96.89 | 87.72 | 92.00 | 75.00 | 94.79 | 72.73 |
| | cognitive | 80.13 | 64.71 | 63.10 | 35.35 | 65.95 | 60.00 | 46.25 | 0.00 | 82.40 | 46.27 | 84.88 | 64.91 | 82.79 | 39.06 | 45.49 | 13.64 |
| Yi-1.5-9B | random | 86.83 | 79.41 | 82.02 | 54.55 | 77.40 | 35.00 | 81.53 | 60.00 | 76.04 | 36.32 | 89.83 | 36.84 | 87.53 | 42.19 | 86.27 | 63.64 |
| | cognitive | 52.76 | 21.76 | 45.99 | 9.09 | 47.25 | 2.50 | 48.10 | 40.00 | 54.22 | 16.92 | 52.41 | 15.79 | 82.75 | 26.56 | 62.85 | 18.18 |
| Yi-1.5-6B | random | 80.64 | 69.41 | 68.82 | 38.38 | 77.83 | 55.00 | 69.61 | 60.00 | 73.33 | 43.78 | 77.71 | 22.81 | 81.65 | 29.69 | 88.54 | 72.73 |
| | cognitive | 49.90 | 15.29 | 68.23 | 41.41 | 49.54 | 2.50 | 42.92 | 0.00 | 76.64 | 43.78 | 68.53 | 14.04 | 44.94 | 0.00 | 86.28 | 50.00 |

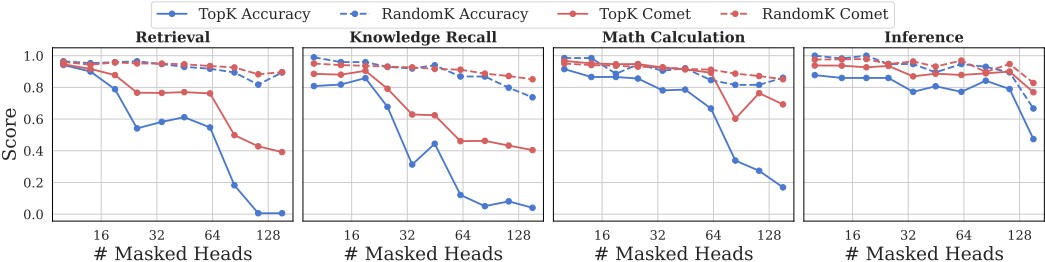

Figure 3: The performance of Llama3.1-8B-instruct by masking out top K cognitive heads vs K random heads on retrieval, knowledge recall, math calculation, and inference.

**Sparsity and Universality:** As shown in Figure 2, each cognitive function activates only a small number of high-importance attention heads, revealing a strikingly sparse pattern. In Llama3.1-8B-instruct, fewer than 7% of all heads have importance scores above 0.001 across the eight functions, suggesting that only a compact subset of heads meaningfully contribute to task performance. This sparsity is not uniform: Retrieval contains the highest proportion of salient heads (6.45% exceeding 0.01), while Inference has the fewest (3.42%). These results highlight that LLMs rely on highly specialized, localized components for different cognitive abilities. Importantly, we observe that this sparse functional organization is consistent across different model architectures and sizes. Additional heatmaps for five other models are provided in Appendix A.1, supporting the universality of this phenomenon.

**Layered Functional Organization:** In addition to sparsity, attention heads show a structured distribution across model layers. Retrieval-related heads cluster primarily in the middle layers, while math-related heads appear more frequently in higher layers. This structured, task-dependent localization points to an emergent modular organization, where different layers support distinct cognitive operations. Further, we identify cognitive heads by selecting those before the elbow point of each function's descending importance curve (Appendix A.2), and find notable variation in head counts across functions (Appendix A.8). For example, in the LLaMA family, mathematical calculation requires fewer heads (59 in Llama3.1-8B-Instruct, 35 in Llama3.2-3B-Instruct), while inference draws on substantially more (139 and 98, respectively), reflecting differences in representational and computational complexity.

## 4.2 Functional Contributions of Cognitive Heads

After identifying the cognitive heads associated with each function, we examine their functional roles by evaluating the model's behavior on the CogQA test set under targeted interventions. We perform

Table 2: Intervention results (%) of different cognitive heads and random heads across Retrieval and Knowledge Recall functions.

| Model | Inter_Head | Retrieval (comet) | Retrieval (acc) | Recall (comet) | Recall (acc) |
|---|---|---|---|---|---|
| Llama3.1-8B | random | 90.83 | 84.71 | 87.85 | 83.84 |
| Llama3.1-8B | retrieval | **44.96** | **8.24** | 72.05 | **33.33** |
| Llama3.1-8B | recall | 86.79 | 75.29 | **56.93** | 38.38 |
| Qwen3-8B | random | 92.81 | 75.29 | 89.90 | 53.54 |
| Qwen3-8B | retrieval | **59.19** | **38.24** | 79.26 | 57.58 |
| Qwen3-8B | recall | 83.31 | 71.18 | **64.81** | **30.30** |

head ablation by scaling the output of a specific attention head with a small factor $\epsilon$ (e.g., 0.001), effectively suppressing its contribution:

$$x_i^{\mathrm{mask}} = \mathrm{Softmax}\left(\frac{W_q^i W_k^{iT}}{\sqrt{d_k/n}}\right) \cdot \epsilon W_v^i \qquad (3)$$

Specifically, we compare model performance when masking identified cognitive heads versus masking an equal number of randomly selected heads. To quantify the impact of masking, we use several standard evaluation metrics including COMET [27], BLEU [26], ROUGE [9], and semantic similarity to compare the model's outputs before and after intervention. We define an output as unaffected if the BLEU score exceeds 0.8, or either the ROUGE or semantic similarity scores surpass 0.6, and compute accuracy accordingly.

As shown in Table 1, masking cognitive heads leads to a significant decline in performance, whereas masking an equal number of random heads results in only marginal degradation across all LLMs. In some cases, masking the identified cognitive heads causes the accuracy to drop to zero, indicating that the model cannot execute the corresponding function without them. This sharp contrast highlights the essential role cognitive heads play in enabling specific reasoning capabilities. To further validate the functional specialization, we conduct experiments where we mask the retrieval heads during the evaluation of knowledge recall (Recall), and conversely, mask knowledge recall heads during the evaluation of retrieval performance. The results in Table 2 show that masking the corresponding cognitive heads causes a significantly larger performance drop than masking others.

We further investigate the performance of model under different numbers of masked attention heads. As shown in Figure 3, increasing the number of randomly masked heads has minimal impact on overall performance of Llama3.1-8B-instruct. In contrast, masking cognitive heads results in a significant drop in performance across various functions. Notably, masking heads associated with Retrieval and Knowledge Recall causes a pronounced degradation in their respective functions, whereas functions such as Math Calculation and Inference exhibit more resilience. This suggests that certain cognitive functions depend more heavily on specific, distinguishable attention heads, while others are distributed more broadly across the model.

## 4.3 Relationship Among Cognitive Heads

While cognitive heads are specialized for distinct functions, understanding their relationships is crucial for revealing how complex reasoning emerges from their cooperation.

**Functional Clustering:** Inspired by neuroscience findings that related cognitive functions localize in overlapping brain regions (e.g., prefrontal cortex for reasoning and inference [5]), we investigate whether LLM attention heads show similar patterns. We rank each head's importance across eight cognitive functions, form ranking vectors, and apply principal component analysis (PCA) to visualize their organization (Figure 4). The results reveal clear clustering: heads linked to reasoning, inference, and decision-making group closely, while those related to mathematical computation form a distinct cluster in Llama and Qwen, and lie adjacent to reasoning heads in Yi. Lower-level functions also show moderate clustering. These patterns suggest a modular functional architecture in LLMs akin to that in the human brain.

**Hierarchical Structure:** Human problem solving often involves hierarchical reasoning, where lower-level functions such as retrieval and comprehension support higher-level inference and decision-

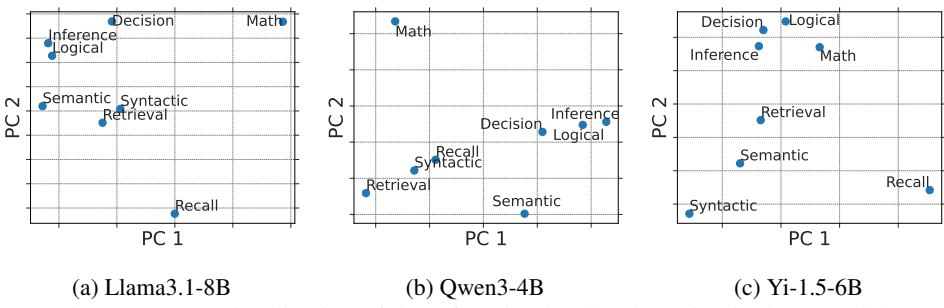

(a) Llama3.1-8B    (b) Qwen3-4B    (c) Yi-1.5-6B

Figure 4: PCA visualization of the 8 function heads' clustering in three models.

Table 3: Study on the influence of low-level cognitive heads for high-order function on Llama3.1-8B-instruct. Accuracy is measured based on BLEU, ROUGE, and semantic similarity scores.

| Retrieval | Knowledge | Semantic | Syntactic | Math | Inference | Logical | Decision |
|:---:|:---:|:---:|:---:|:---:|:---:|:---:|:---:|
| ✗ | ✓ | ✓ | ✓ | 0.00 ↓ 100 | 0.00 ↓ 100 | 0.00 ↓ 100 | 0.00 ↓ 100 |
| ✓ | ✗ | ✓ | ✓ | 0.00 ↓ 100 | 0.00 ↓ 100 | 0.00 ↓ 100 | 0.00 ↓ 100 |
| ✓ | ✓ | ✗ | ✓ | 66.67 ↓ 33.33 | 88.24 ↓ 11.76 | 93.10 ↓ 8.90 | 57.14 ↓ 42.86 |
| ✓ | ✓ | ✓ | ✗ | - | 76.92 ↓ 23.08 | 100 0.00 | 100 0.00 |

making. The CogQA dataset captures this structure through subquestions progressing from simple information extraction to complex reasoning. We test if LLMs reflect this hierarchy by masking attention heads tied to early-stage functions and measuring the effect on later tasks. For instance, to assess how Retrieval affects Math Calculation, we suppress Retrieval-related heads throughout the subquestions. Answers from earlier Retrieval are used as priors for later math reasoning, allowing us to observe how disrupting low-level functions can propagate and impair higher-level reasoning along the chain. As Table 3 shows, masking retrieval or knowledge recall heads causes significant performance drops in subsequent decision-making steps, whereas masking syntactic understanding heads has minimal impact. This provides evidence for an emergent hierarchical organization in LLMs, where foundational cognitive functions underpin advanced reasoning.

## 4.4 Influence of Cognitive Heads on Downstream Tasks

In this section, we investigate how cognitive heads influence downstream tasks through both negative interventions (masking out cognitive function heads) and positive interventions (shifting heads toward specific functions). We conduct experiments on two tasks: a math task using 100 GSM8K samples (GSM8K_100) and a retrieval task with 49 samples from an extractive_QA dataset. The Extractive_QA pairs are generated by GPT-4o, with answers extracted directly from the source paragraph.

**Negative Intervention:** We perform negative intervention by masking corresponding cognitive heads (Math Calculation heads for GSM8K_100 and Retrieval heads for Extractive_QA), effectively suppressing their activations. As shown in Table 4, this causes significant performance drops across models, confirming these heads' functional roles. Notably, after masking, performance converges to a similarly low level across different LLMs, regardless of model size or original accuracy. This is expected, as the crucial cognitive heads responsible for specific functions are disabled, making it difficult for the model to arrive at correct answers.

For math, the remaining 30% accuracy likely stems from two factors: (1) memorized answers in the base model, and (2) simple questions not requiring actual computation. For retrieval, masking Retrieval heads almost completely abolishes the model's retrieval ability across all scales. This indicates that cognitive functions are indeed localized in a subset of heads, and masking them leads to a systematic degradation, irrespective of model capacity. The negative intervention example further shows that, masking the **Math Calculation** heads leads to errors in arithmetic tasks, while retrieval and language functions remain largely unaffected. This confirms that these cognitive heads are crucial for specific functions and highlights the robustness and generalizability of our method.

**Positive Intervention:** We calculate the activation directions of different cognitive functions using the CogQA dataset. For each function, the activation direction of a head at layer $l$ and index $h$ is computed as:

$$\mathrm{dir}_l^h = \mathbb{E}_{i \in \mathcal{D}_{\text{correct}}} \left[ x_l^h(i) \right] - \mathbb{E}_{i \in \mathcal{D}_{\text{incorrect}}} \left[ x_l^h(i) \right] \tag{4}$$

where $x_l^h(i)$ denotes the activation of head at layer $l$ and index $h$, and $\mathcal{D}_{\text{correct}}$ and $\mathcal{D}_{\text{incorrect}}$ represent the sets of samples answered correctly and incorrectly, respectively. Then we estimate the standard deviation of activations [17] along the cognitive function direction to be $\sigma_l^h$, and shift original head activation as $x_l^h(i) \leftarrow x_l^h(i) + \alpha \sigma_l^h \, \mathrm{dir}_l^h$, where $\alpha$ is a parameter.

The experimental results in Table 4 show that enhancing the activation of retrieval heads along their corresponding functional directions leads to improved performance on the retrieval task. For the math task, questions often engage multiple cognitive functions. We observe cases where computations are correct but semantic understanding fails. By positively intervening on either Math Calculation or Semantic Understanding heads, performance consistently improves, highlighting the collaborative nature of cognitive functions in complex reasoning and the need for joint interventions—a direction we leave for future work.

The example below illustrates a positive intervention on **Math Calculation** heads in Qwen3-4B: the model initially produces an incorrect answer (red) but generates the correct one (blue) after intervention. This demonstrates that adjusting cognitive head activations along their functional directions reliably enhances performance on complex reasoning tasks.

---

**Negative Intervention Case**

**Query:** Question: Solve -42*r + 27*c = -1167 and 130*r + 4*c = 372 for r.
**Original Output:** Okay, so I need to solve this system of equations: -42r + 27c = -1167 and 130r + 4c = 372 ... First equation: -42*4 + 27*(-37) = -168 - 999 = -1167. Correct. Second equation: 130*4 + 4*(-37) = 520 - 148 = 372. Correct. Yes, that's right. So everything checks out. So I think that's the correct answer. Therefore, r is 4.
**Negative Intervention (Math) Output:** ... To eliminate $c$, I need the coefficients of $c$ in both equations to be the same (or negatives). The coefficients are 27 and 4. The least common multiple of 27 and 4 is 100? Wait, 27 is $3^3$, and 4 is $2^2$, so LCM is $2^2 * 3^3 = 4 * 27 = 100$. So if I can make the coefficients of $c$ to be 100 and 100, or 27 and 4. Wait, but 27 and 4. So if I multiply the first equation by 4 and the second equation by 27, then the coefficients of $c$ would be $27*4 = 100$ and $4*27 = 100$. Then, if I subtract the equations, the $c$ terms would be 100 and 100, but since they are both positive, subtracting the equations would eliminate $c$. ...

---

**Positive Intervention Case**

**Query:** On a school trip to the seashore, Alan and his friends collected shells. Alan collected four times as many shells as Ben did. Ben got a late start and only collected a third of what Laurie did. If Laurie collected 36 shells how many did Alan collect?
**Original Output:** Alan collected four times as many shells as Ben did, and Ben collected a third of what Laurie did. Since Laurie collected 36 shells, Ben collected 36 / 3 = 12 shells. Therefore, Alan collected 4 * 12 = 48 shells. However, the question asks for the number of shells Alan collected, and it is given that Alan collected four times as many shells as Ben did. Therefore, the correct answer is 4 * 12 = 48 * 4 / 3 = 72.
**Positive Intervention (Math) Output :** ...Since Ben collected a third of what Laurie did, he collected 36 / 3 = 12 shells. Alan collected four times as many shells as Ben did, so he collected 4 * 12 = 48 shells...The correct answer is 48

---

## 5 Related Works

**Neural Networks and the Brain**   Neural networks have long been studied as computational models of the brain, with early work linking artificial neurons to the biological mechanisms of perception and learning [19]. Convolutional neural networks (CNNs), in particular, have been shown to capture representations similar to those in the visual cortex [37], and more recent studies suggest that the functional modularity observed in deep networks gives rise to brain-like specialization [13] in vision

Table 4: The LLM performance on GSM8k_100 and Extractive_QA by Negative and Positive Intervention (Inter.). Math accuracy (%) is measured by exact answer match; Extractive_QA accuracy checks if the original paragraph answer appears in the response.

| Dataset | Method | Head | Llama3.1-8B | Llama3.2-3B | Qwen3-8B | Qwen3-4B |
|---------|--------|------|-------------|-------------|----------|----------|
| Extractive_QA | Base | - | 57.14 | 36.73 | 57.14 | 51.02 |
| | Negative Inter. | Retrieval | 0 | 0 | 14.29 | 12.24 |
| | Positive Inter. | Retrieval | 63.26 | 44.90 | 61.22 | 69.38 |
| GSM8K_100 | Base | - | 82 | 64 | 94 | 91 |
| | Negative Inter. | Math | 38 | 34 | 34 | 37 |
| | Positive Inter. | Math | 84 | 66 | 94 | 92 |
| | Positive Inter. | Semantic | 84 | 65 | 94 | 93 |

task. More recently, LLMs have exhibited striking parallels with human brain activity during language processing. In particular, transformer-based models, such as GPT-2, produce internal representations that align with neural responses in language-selective brain regions [8, 28]. However, prior work mostly focuses on perception and language representations, with limited study on higher-level cognitive functions like reasoning. We instead analyze LLMs' behavior in complex reasoning tasks to explore their alignment with human cognitive functions and functional specialization.

**Functional Specialization of Attention Heads**    Recent years have witnessed growing interest in understanding the functional roles of attention heads in Transformer-based models, forming a core component of mechanistic interpretability research. Early work by [10] demonstrated that individual heads in BERT capture specific linguistic phenomena such as syntactic dependencies and coreference, indicating a degree of functional specialization. Building on this, [32] proposed a pruning-based approach to identify important heads by measuring their contribution to downstream performance, showing that many heads are redundant. Subsequent studies extended this analysis to decoder-only large language models (LLMs). [21] explored functional decomposition in such models, leading to the identification of distinct attention heads responsible for tasks such as pattern induction [22], truthfulness [17], information retrieval [36], and safety alignment [41]. For a broader survey, see [40]. Despite these advances, most prior work focuses on isolated heads and evaluates them in relatively simple or synthetic tasks. In contrast, we investigate functionally specialized heads under more complex reasoning settings by aligning attention head behavior with human cognitive functions.

# 6    Limitations and Future works

While our study provides an initial framework for analyzing the cognitive functions of attention heads, several limitations remain. We examine only eight predefined functions, which, though representative, may not cover the full range of LLM capabilities; future work could expand this taxonomy with finer-grained functions. Each CogQA subquestion is labeled with a single cognitive function, though real reasoning may involve multiple functions. Likewise, we assume one head corresponds to one function, even though a head may support multiple or context-dependent roles. These complexities are not fully addressed in our current framework. Excluding subquestions with incorrect subanswers could improve multi-class probing, and further investigation is needed to understand heads serving multiple functions. While our study focuses on analysis rather than application, identifying cognitively relevant heads could inform model design, including dynamic head activation, improved chain-of-thought prompting, targeted fine-tuning, or modular architectures—directions we leave for future exploration.

# 7    Conclusions

We propose an interpretability framework that connects attention heads in large language models (LLMs) to human cognitive functions involved in reasoning. To support this, we introduce CogQA, a cognitively grounded dataset, along with a multi-class classification approach to identify specialized heads associated with specific reasoning tasks. Our analysis across multiple LLM families and scales demonstrates that attention heads exhibit universality, sparsity, intrinsic roles, and dynamic, hierarchical organization. These findings indicate that LLMs internally organize reasoning processes in a manner akin to human cognition, laying the groundwork for more interpretable and cognitively informed language models.

## Acknowledgments

This work is partially supported by the following Australian Research Council (ARC) projects: FT220100318, DP220102121, LP220100527, LP220200949, DP230101534.

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

# A Appendix

## A.1 The cognitive function distribution of other models

We present the heatmaps for the remaining five models in this section. The results reveal a notable universality in the sparsity patterns of attention heads across different architectures. Moreover, models within the same family tend to exhibit similar sparsity distributions. For instance, Llama3.2-3B (Figure 5) and Llama3.1-8B (Figure 2) share comparable patterns, as do Qwen3-4B (Figure 7) and Qwen3-8B (Figure 6), as well as Yi-1.5-6B (Figure 9) and Yi-1.5-9B (Figure 8). This consistency is likely due to the shared architectural design and similar pretraining data within each model family.

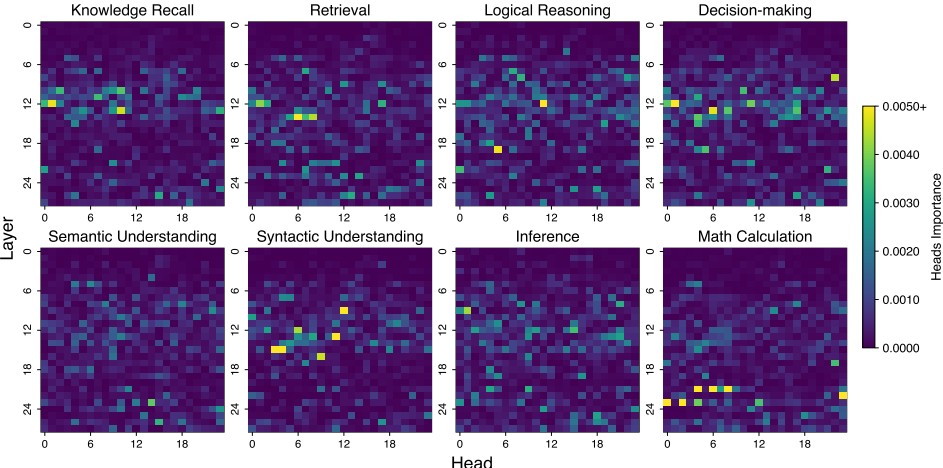

Figure 5: The existence of cognitive heads in Llama3.2-3B-instruct responsible for eight distinct functions in complex reasoning tasks. The x-axis represents the head index, while the y-axis indicates the layer index.

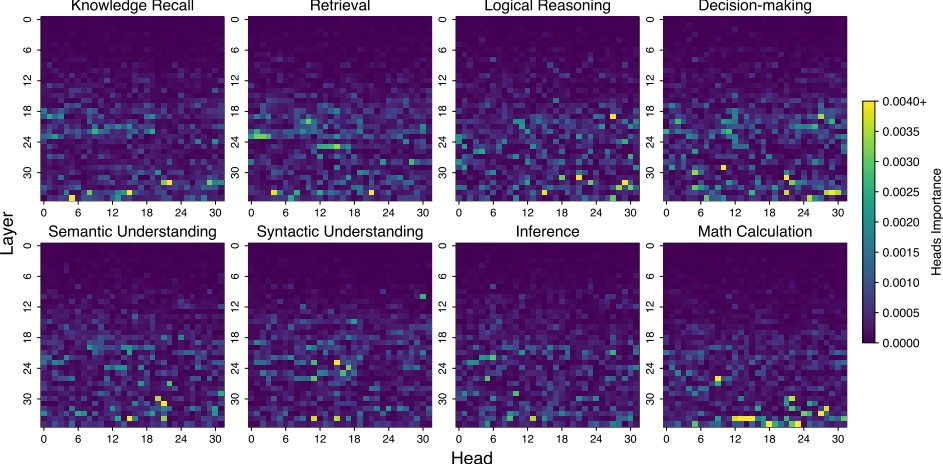

Figure 6: The existence of cognitive heads in Qwen3-8B responsible for eight distinct functions in complex reasoning tasks. The x-axis represents the head index, while the y-axis indicates the layer index.

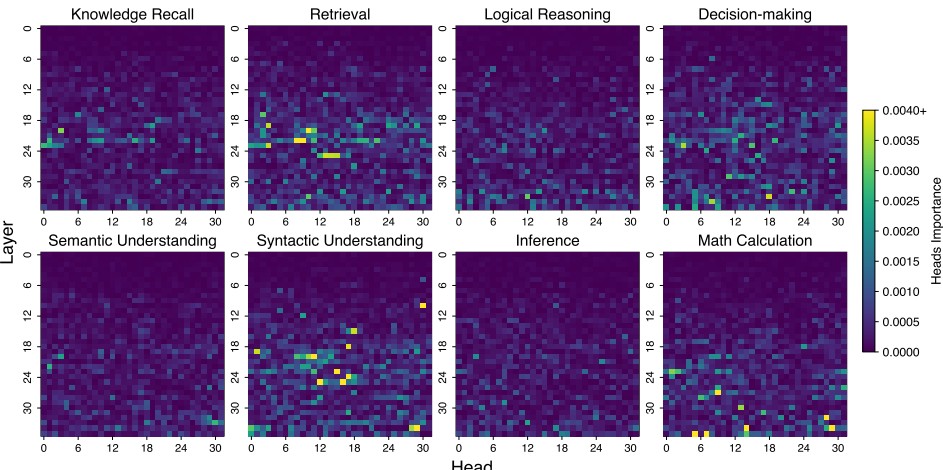

Figure 7: The existence of cognitive heads in Qwen3-4B responsible for eight distinct functions in complex reasoning tasks. The x-axis represents the head index, while the y-axis indicates the layer index.

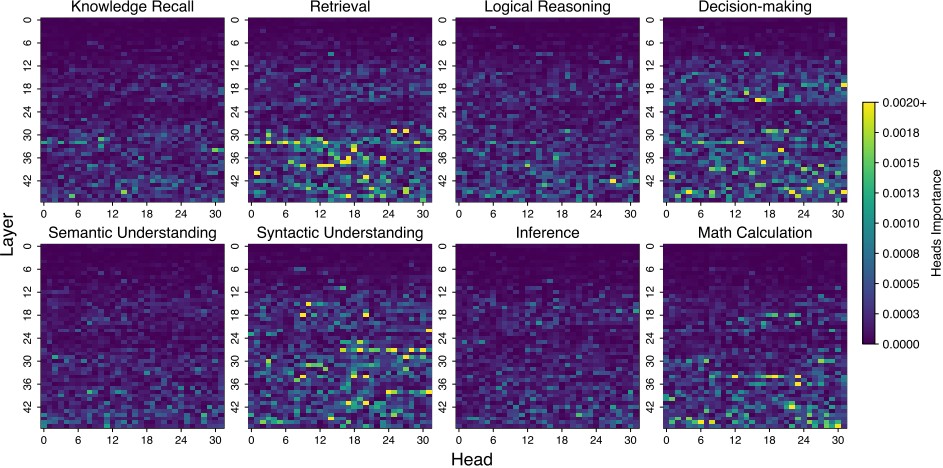

Figure 8: The existence of cognitive heads in Yi-1.5-9B responsible for eight distinct functions in complex reasoning tasks. The x-axis represents the head index, while the y-axis indicates the layer index.

## A.2 Importance curve

We ranked the importance scores and identified the elbow point, as illustrated in Figure 10.

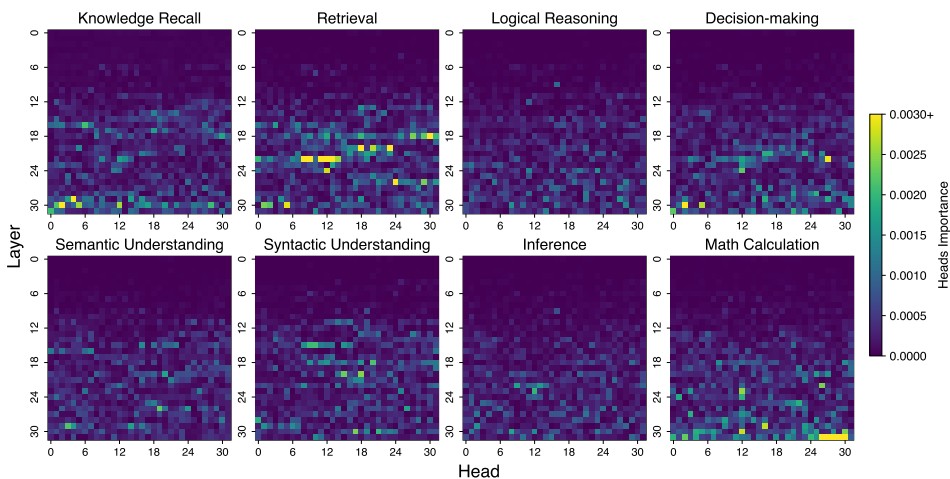

Figure 9: The existence of cognitive heads in Yi-1.5-6B responsible for eight distinct functions in complex reasoning tasks. The x-axis represents the head index, while the y-axis indicates the layer index.

Table 5: The test accuracy (%) of probing method on different LLMs.

| Dataset | Llama3.1-8B-instruct | Llama3.2-3B-instruct | Qwen3-8B | Qwen3-4B | Yi-1.5-9B | Yi-1.5-6B |
|---------|---------------------|---------------------|----------|----------|-----------|-----------|
| CogQA | 83.73 | 79.80 | 84.71 | 80.79 | 77.56 | 75.18 |

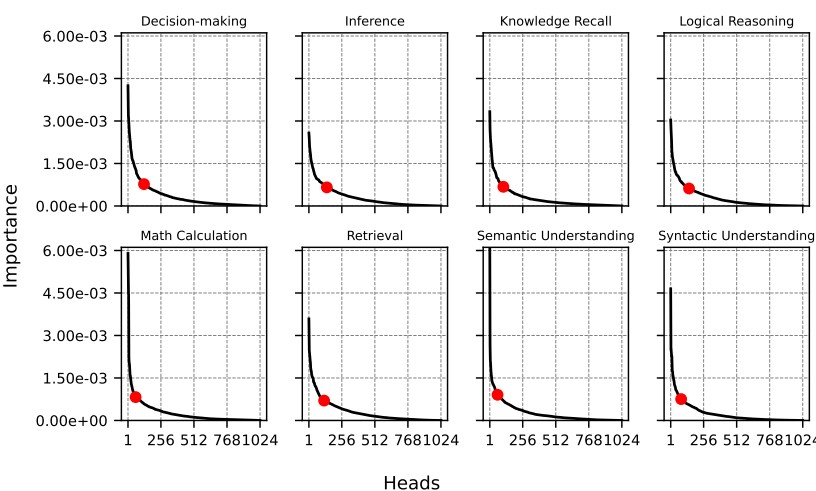

Figure 10: Importance curve for eight functions, Llama3.1-8B-instruct.

## A.3   MLP

We train a two-layer multi-class MLP for cognitive function classification. The first layer applies a shared linear projection to each multi-head representation vector, reducing each to a 64-dimensional embedding. These embeddings are then flattened and concatenated into a single vector of size $64 \times number of heads$. This vector is fed into a hidden layer with 512 units, followed by a ReLU activation and a dropout with a rate of 0.3. The final output layer maps the 512-dimensional hidden representation to the set of cognitive function labels.

The model is trained using the Adam optimizer with a learning rate of $10^{-4}$ and a cross-entropy loss. Training proceeds for 100 epochs. The test accuracy of our classification method across all LLM models is summarized in the Table 5.

## A.4 Prompt for Generating CogQA

---

**Prompt**

**Prompt:** You are an expert in analytical logical reasoning. You will be given a question along with its chain-of-thought process. Your task is to break the question down into subquestions based on the chain-of-thought process, ensuring that all necessary steps for solving the problem and constructing the logical chain are included to simulate critical thinking.

Decompose the Question: Identify and formulate the key subquestions required to solve the main question logically. Fill in Missing Steps: Ensure that all essential reasoning steps are explicitly stated.

NOTE: The information of chain-of-thought cannot be used directly if it doesn't exist in main query. Each subquestion should be derived solely from the main query and the preceding subquestion. Answer the Subquestions: Provide clear, step-by-step solutions for each subquestion. Annotate Cognitive Skills: Identify and label the specific cognitive abilities required to answer each subquestion. If you believe other cognitive skills are relevant, you may also consider incorporating them. You will be given predefined labels along with their descriptions. Your goal is to enhance the logical reasoning process by making it explicit and structured.

<cognitive_skills> **Retrieval**: Refers to the process of fetching relevant information from input text, typically involving the extraction of specific words, phrases, or sentences directly from the original text. **Knowledge Recall**: Involves the storage and recall of domain-specific knowledge, such as concepts from math, physics, biology, etc. This is typically the internal knowledge base of a language model. (Corresponding to the memory head) **Semantic Understanding**: Refers to the ability to comprehend and extract meaning from text or symbols by recognizing relationships between words, phrases, and concepts. It goes beyond syntactic understanding by grasping context, intent, and underlying knowledge. **Syntactic Understanding***: Involves the ability to analyze and interpret the grammatical structure of sentences, including the roles and relationships of words, phrases, and clauses within the language. **Math Calculation**: Refers to the process of performing arithmetic or mathematical operations to obtain a result. It involves applying mathematical concepts, such as addition, subtraction, multiplication, division, and more complex operations (e.g., algebra, calculus), to solve problems or derive values from given inputs. **Inference**: Involves drawing conclusions based on existing evidence or information. It follows logical rules to deduce new statements or decisions from given information. **Logical Reasoning**: The process of drawing conclusions based on a set of premises, following established rules of logic, used to ensure that decisions of people are coherent, consistent, and based on sound principles. **Decision-making**: The process of making a choice in a selection task based on previous information or analysis. <cognitive_skills>

Here is the question: <question> question <question>

Here is the chain-of-thought: <chain-of-thought> cot <chain-of-thought>

Note

- Your task is to break the question down into detailed subquestions, ensuring each subquestion can be answered using only one specific cognitive skill. - You need to create a structured and explicit reasoning process that simulates critical thinking while maintaining clarity and precision. - The subquestion needs to be easy to answer and the answer needs to be concise

- The information of chain-of-thought cannot be used directly if it doesn't exist in main query. - Each subquestion should be derived solely from the main query and the preceding subquestion. - You CAN NOT retrieval information from chain-of-thought, but you can retrieval from question. - Your output should be formatted as a list of JSON objects, where each object represents a subquestion, its answer, and the required cognitive skill. - You should use the most efficient logic to analyze the problem and minimize the number of subquestions.

Output format [ "subquestion": "<Subquestion text>", "answer": "<Concise answer>", "cognitive_skill": "<Assigned cognitive skill>" , "subquestion": "<Subquestion text>", "answer": "<Concise answer>", "cognitive_skill": "<Assigned cognitive skill>" ]

Your answer:

---

## A.5 Annotations

To ensure the quality and reliability of the decomposed subQAC triplets in the CogQA dataset, we design a rigorous multi-stage annotation pipeline, combining expert review and model-based verification. The goal is to verify the logical validity of subquestions, the correctness of their associated cognitive function labels, and the accuracy of the answers.

**Stage 1: Validating Subquestion Decomposition**   In the first stage, we evaluate whether the generated subquestions are logically sound and align with natural human reasoning. For each QA pair, three expert annotators (with backgrounds in linguistics or cognitive science) independently assess the validity of each subquestion. A subquestion is marked `true` if it meaningfully contributes to answering the main question and follows a logical reasoning trajectory. Otherwise, it is marked `false`.

If a subquestion depends on prior information—such as the question text or the answer—from another subquestion, the subquestion order must reflect this dependency. While some subquestions can be answered in parallel and are order-independent, others have prerequisite relationships that require a specific sequence. As the overall reasoning structure often forms a graph, where both sequential and parallel dependencies coexist. During LLM inference, we include the previous subquestions and their corresponding subanswers in the prompt as prior information. Thus, the critical factor is not the ordering alone, but whether the prompt provides the necessary context to answer the current subquestion accurately.

We apply the following filtering criteria:

- **AI-Human Agreement**: If any annotator considers fewer than 60% of the subquestions valid, the entire QA decomposition is discarded.
- **Inter-Annotator Agreement**: A subquestion is deemed invalid if at least two annotators mark it as `false`. If over 40% of the subquestions in a QA pair are invalid under this rule, the whole QA pair is removed.

This filtering ensures that the retained QA decompositions follow coherent, cognitively plausible reasoning chains.

**Stage 2: Verifying Cognitive Function Labels**   In the second stage, annotators evaluate the correctness of the cognitive function label $c_i$ assigned to each subQAC triplet $(q_i, a_i, c_i)$. Three annotators independently mark each label as `true` or `false`. When discrepancies occur, annotators collaboratively reassign the correct cognitive label to ensure alignment with the underlying mental operation.

This step ensures that the categorization of subquestions accurately reflects established distinctions between information retrieval, semantic understanding, logical reasoning, and other cognitive processes.

**Stage 3: Answer Verification via Model and Human Review**   In the final stage, we verify the correctness of each answer $a_i$ using both automated and manual procedures. We employ the o4-mini model [25], known for its logical reasoning capabilities, to re-evaluate GPT-4o-generated answers. If o4-mini disagrees with GPT-4o, it provides an alternative answer. A human annotator then compares both answers and resolves discrepancies by supplying the correct one when necessary. Given the generally objective nature of answers, only one annotator is required for this task.

**Annotation Outcome**   Following this multi-stage process, we retain 570 validated QA pairs, yielding a total of 3,402 high-quality subQAC triplets. Notably, we augment certain cognitive functions to ensure balance across categories. As a result, the original 570 QA pairs were expanded to 720 (including some duplicates), with each duplicated pair potentially associated with distinct subquestions and cognitive functions.

## A.6 CogQA Example

Table 6 presents illustrative examples from the CogQA dataset. The main question and its corresponding answer are taken from the original dataset. Based on an analysis of the main question, a sequence of sub-questions, their answers, and associated cognitive function labels are generated in order.

Table 6: Two examples from the CogQA dataset showing a main question, its final answer, and a breakdown into subquestions with answers and their corresponding cognitive function labels.

**Example 1:**

| Main Question | A one-year subscription to a newspaper is offered with a 45% discount. How much does the discounted subscription cost if a subscription normally costs $80? |
|---|---|
| Answer | We calculate first the discount: $80 \times 45 / 100$ = $36. So, the discounted subscription amounts to 80 – 36 = $44. |

| Subquestion | Answer | Cognitive Label |
|---|---|---|
| 1. What is the normal cost of a one-year subscription to the newspaper? | $80 | Retrieval |
| 2. What is the discount percentage offered on the subscription? | 45% | Retrieval |
| 3. How much is the discount amount in dollars for the subscription? | $36 | Math Calculation |
| 4. What is the cost of the subscription after applying the discount? | $44 | Math Calculation |

**Example 2:**

| Main Question | What does every person talk out of? Options: - name - hide - mother and father - mouth - heart |
|---|---|
| Answer | By mouth, talking is done. Every person talk out of mouth. |

| Subquestion | Answer | Cognitive Label |
|---|---|---|
| 1. What is the primary function of talking? | To communicate verbally. | Knowledge Recall |
| 2. Which part of the human body is primarily used for verbal communication? | Mouth | Knowledge Recall |
| 3. Based on the options provided, which option corresponds to the part used for verbal communication? | Mouth | Decision-making |

## A.7 Prompt for Question Asking

> **Prompt**
>
> **Prompt:** You are an expert in analytical and logical reasoning. You will be given a main question and prior knowledge in chain-of-thought (CoT) format. Your task is to answer a follow-up subquestion using the information provided.
> Here is the main question:
> <main_question> question </main_question>
> Here is the prior knowledge in chain-of-thought (CoT) format:
> <prior_knowledge> cot </prior_knowledge>
> Here is the subquestion:
> <subquestion> subquestion </subquestion>
> Instructions:
> - Answer the subquestion carefully.
> - You can use the information in the prior_knowledge to help you answer the subquestion.
> - Your response should be clear and concise.
> - Stick to factual reasoning based on provided CoT.
> - Do not include any explanation, commentary, or code.
> - Do not output anything after the closing square bracket ']'.
> Only output your final answer using this format: [ "answer": "<Your answer here>" ]
> Your answer:

Table 7: Count (C) and percentage (%) of attention heads exceeding elbow thresholds for each cognitive function across six models.

| Model | Recall | | Retrieval | | Logical | | Decision | | Semantic | | Syntactic | | Inference | | Math | |
|---|---|---|---|---|---|---|---|---|---|---|---|---|---|---|---|---|
| | C | % | C | % | C | % | C | % | C | % | C | % | C | % | C | % |
| Llama3.1-8B-instruct | 105 | 10.3 | 118 | 11.5 | 142 | 13.9 | 124 | 12.1 | 60 | 5.9 | 81 | 7.9 | 139 | 13.6 | 59 | 5.8 |
| Llama3.2-3B-instruct | 95 | 14.1 | 62 | 9.2 | 95 | 14.1 | 87 | 12.9 | 90 | 13.4 | 63 | 9.4 | 98 | 14.6 | 35 | 5.2 |
| Qwen3-8B | 119 | 10.3 | 115 | 10.0 | 114 | 9.9 | 87 | 7.6 | 68 | 5.9 | 108 | 9.4 | 178 | 15.5 | 61 | 5.3 |
| Qwen3-4B | 115 | 10.0 | 94 | 8.2 | 120 | 10.4 | 170 | 14.8 | 143 | 12.4 | 106 | 9.2 | 109 | 9.5 | 99 | 8.6 |
| Yi-1.5-9B | 200 | 13.0 | 134 | 8.7 | 134 | 8.7 | 174 | 11.3 | 218 | 14.2 | 140 | 9.1 | 173 | 11.3 | 167 | 10.9 |
| Yi-1.5-6B | 118 | 11.5 | 90 | 8.8 | 200 | 19.5 | 93 | 9.1 | 99 | 9.7 | 142 | 13.9 | 146 | 14.3 | 67 | 6.5 |

## A.8   The number of cognitive heads for different LLMs

The number of cognitive heads for each model is shown in Table 7.

## A.9   Ablation study - Different position of head activation

In the main experiments, we use the top-k generated tokens and average their multi-head attention vectors. We also explore alternative strategies for extracting representations, including using the first generated token, the last generated token, the first meaningful token, and the average of all generated tokens. The corresponding results are shown in Table 8.

Here, first is the first token, last is the last token, meaning_first is the first meaning token (excluding formatting), top-k is the top-k most semantically important tokens, full is all tokens in the answer. We observe that top-k token masking leads to the most significant performance drop when masking the top-30 identified heads, indicating higher precision in identifying retrieval-relevant heads. Interestingly, last, meaning_first, full, and top-k show similar performance trends. This is because different tokens in the output contribute to answering the question, and as the number of masked cognitive heads increases, the influence of token using decreases. Additionally, for Retrieval, the full answer is usually meaningful, whereas others like Math Calculation require semantically meaningful tokens. Based on these results, we choose top-k as our final setting.

Table 8: Attention heads associated with cognitive functions are selected based on different token positions. Accuracy and COMET scores are evaluated after intervention; lower values indicate better outcomes.

| Model | Head_num | Token_use | Retrieval(comet) | Retrieval(acc) | Math(comet) | Math(acc) |
|---|---|---|---|---|---|---|
| Llama3.1-8B | 30 | first | 90.51 | 83.53 | 91.13 | 73.13 |
| Llama3.1-8B | 30 | last | 86.86 | 81.76 | 90.04 | 68.66 |
| Llama3.1-8B | 30 | meaning_first | 88.13 | 79.41 | 89.72 | 68.66 |
| Llama3.1-8B | 30 | full | 73.93 | 47.06 | 89.92 | 69.15 |
| Llama3.1-8B | 30 | top-k | **70.05** | **46.47** | 89.32 | 67.16 |
| Llama3.1-8B | 50 | first | 93.28 | 89.41 | 94.46 | 89.57 |
| Llama3.1-8B | 50 | last | 64.39 | 41.18 | 92.05 | 70.15 |
| Llama3.1-8B | 50 | meaning_first | 62.90 | 34.12 | 84.60 | 60.69 |
| Llama3.1-8B | 50 | full | **46.20** | **11.76** | 89.01 | 78.11 |
| Llama3.1-8B | 50 | top-k | 65.64 | 47.76 | 89.65 | 70.15 |

## A.10   Examples of top-$k$ tokens

The selected tokens are intended to semantically represent the generated answer. Below are examples for different cognitive functions for Llama3.1-8B-instruct:

Table 9: Examples of question decomposition with cognitive functions and token selection.

| Main Question | Subquestion | Cognitive Function | Answer | Selected Tokens |
|---|---|---|---|---|
| Given the sentence "A surfboarder catches the waves." can we conclude that "A surfboarder in the water."? (Options: yes / it is not possible to tell / no) | What is typically required for a surfboarder to catch waves? | Knowledge Recall | The surfboarder needs to be in the water. | ['surfboarder', 'needs', 'be', 'in', 'water'] |
| Is the following a factual statement? "Due to its high density, countries around the world use Palladium to mint coins." (Options: yes / no) | What is the statement in question? | Retrieval | The statement in question is: Due to its high density, countries around the world use Palladium to mint coins. | ['high', 'density', 'Palladium', 'mint', 'coins'] |
| A one-year subscription to a newspaper is offered with a 45% discount. How much does the discounted subscription cost if a subscription normally costs $80? | How much is the discount amount in dollars for the subscription? | Math Calculation | 36 | ['36'] |

We can see that the selected tokens semantically represent the answer. Note that we use all tokens when the number of tokens is fewer than 5.

