# OpenReview forum: "Cognitive Mirrors: Exploring the Diverse Functional Roles of Attention Heads in LLM Reasoning"
_NeurIPS.cc/2025/Conference — NeurIPS 2025 poster_

### Official Review · Reviewer_8YmT · 2025-06-25

**Clarity:** 3
**Significance:** 3
**Originality:** 4
**Rating:** 4
**Confidence:** 4

**Summary:**

The paper presents an analysis on the roles of attention heads in LLMs in reasoning tasks, aided by their introduced dataset called CogQA. It contains complex questions that reduce into subquestions, similar to a chain-of-thought design. They conduct experiments across several model architecture and model sizes, highlighting how LLMs internally organize reasoning processes.

**Questions:**

Questions

- One of the cognitive functions, “Knowledge Recall” is defined as accessing stored factual or procedural knowledge from memory. However, datasets like CREAK from which the proposed dataset is sourced from contains multiple entities which the model may not be aware of, leading to faults in the analysis of this cognitive function. An example of Knowledge Recall from the training data, “What is the density characteristic of Palladium?”. This knowledge may not be known by every model, making it an unfair comparison.

- The authors make the claim of layered functional organization wherein retrieval related heads cluster primarily in the middle layers while math-related heads appear more frequently in higher layers. Although this is apparent in Llama3.1-8B, the same trend is not too apparent in other models like Qwen3-8B and Yi-1.5-9B. Please provide a discussion and potential reasoning for the same.

- Table 1 provides the accuracy change in a cognitive function after suppressing the attention heads for that particular task. However, performing cross task evaluation (i.e., suppressing cognitive heads for task like Retrieval and evaluating on Knowledge Recall) would provide more insights. Although a similar analysis in provided between lower and higher level cognitive functions, intra-level function relations would substantiate the Functional Clustering results presented in Section 4.3.

- Section 4.4 discusses positive and negative interventions. However, no empirical results (e.g. change in model accuracy on Math tasks after positive intervention on Math related attention heads) are provided. Change in accuracy due to positive/negative interventions in attention heads corresponding to specific cognitive tasks would further solidify their analysis.

**Ethical Concerns:**

["NO or VERY MINOR ethics concerns only"]

**Final Justification:**

The authors resolved my concerns regarding the theory behind cognitive functions, change in results due to positive and negative interventions and the effect of interventions w.r.t model size. I would like to maintain my score. This is a technically solid paper, with good evaluation. But similar works have been proposed in the past and the impact of this work is limited to sub-field of LLM reasoning.

**Limitations:**

yes

**Quality:**

3

**Strengths And Weaknesses:**

Strengths:

- The paper is well written and provides a clear flow of thought. In multiple instances, they draw parallels with human cognition.
- The experiment design is well thought of and evaluates their approach across multiple model families and model sizes.

Weaknesses:

- The paper provides no discussion on how the cognitive functions were decided. Moreover, adding examples for each of the cognitive functions would bring more clarity.
- Value of hyperparameter k used in top-k most important tokens selection is missing
- Table 2 doesn’t specify which model is evaluated

---

> ### Author Rebuttal · Authors · 2025-07-31
>
> Dear Reviewer,
>
> Thank you for your thoughtful comments and suggestions. They have been instrumental in improving the clarity and quality of our paper. Below, we provide detailed, point-by-point responses to your concerns.
>
> **Q1. The paper provides no discussion on how the cognitive functions were decided. Moreover, adding examples for each of the cognitive functions would bring more clarity.**
>
> **A.** Inspired by research in cognitive science [1-2] and recent findings on the roles and mechanisms of attention heads in large language models (LLMs) [3], we focus on eight representative cognitive functions grouped into two categories: low-level and high-level functions, each relevant to complex reasoning tasks. To better align with human cognitive processes, we incorporate human annotations to guide the function labeling for subquestions.
>
> We appreciate your suggestion regarding the addition of examples. To improve clarity, we will include representative examples for each cognitive function in the Appendix. Furthermore, the complete CogQA dataset—with all annotated subquestions—is provided in the supplementary materials for reference.
>
> [1] John R Anderson. Rules of the mind. Psychology Press, 2014.
>
> [2] Adele Diamond. Executive functions. Annual review of psychology, 64(1):135–168, 2013
>
> [3] Zheng, Z., et al. "Attention heads of large language models: A survey. arXiv 2024." arXiv preprint arXiv:2409.03752.
>
> **Q2. Value of hyperparameter k used in top-k most important tokens selection is missing.**
>
> **A.** In our experiments, we set k = 5. This value was chosen to balance capturing enough key information while minimizing the inclusion of less relevant or noisy tokens. We will clarify this setting in the revised manuscript.
>
> **Q3. Table 2 doesn’t specify which model is evaluated.**
>
> **A.** Thank you for pointing this out. Table 2 evaluates on llama3.1\_8B\_instruct model. We will carefully review and clarify this throughout the final version.
>
> **Q4. One of the cognitive functions, “Knowledge Recall” is defined as accessing stored factual or procedural knowledge from memory. However, datasets like CREAK from which the proposed dataset is sourced from contains multiple entities which the model may not be aware of, leading to faults in the analysis of this cognitive function. An example of Knowledge Recall from the training data, “What is the density characteristic of Palladium?”. This knowledge may not be known by every model, making it an unfair comparison.**
>
> **A.** Our goal is to explore the functional roles of attention heads rather than to evaluate whether a model possesses specific factual knowledge. For example, when asked, “What is the density characteristic of Palladium?”, even if the model does not know the answer, it still engages in the process of attempting to recall factual knowledge—much like a human trying to retrieve uncertain information. This process inherently activates attention heads related to the Knowledge Recall function. Of course, incorrect answers may correspond to insignificant cognitive head activation, so the straightforward approach is to exclude them from the analysis. However, according to our observations, such cases are relatively rare. Overall, our analysis focuses on identifying the functional roles involved in the cognitive process and does not involve comparisons between different LLMs.
>
> **Q5. The authors make the claim of layered functional organization wherein retrieval related heads cluster primarily in the middle layers while math-related heads appear more frequently in higher layers. Although this is apparent in Llama3.1-8B, the same trend is not too apparent in other models like Qwen3-8B and Yi-1.5-9B. Please provide a discussion and potential reasoning for the same.**
>
> **A.** As shown in Fig. 6 for Qwen3-8B and Fig. 8 for Yi-1.5-9B, retrieval-related heads are predominantly located in the middle layers, with some also appearing in the deeper layers. This variation may stem from differences in pretraining strategies and data across LLMs. Importantly, our main claim is not that the exact layer locations of functional heads are fixed across models, but rather that cognitive functions tend to exhibit layered preferences, which can be consistently observed across architectures.
>
> **Q6.  Table 1 provides the accuracy change in a cognitive function after suppressing the attention heads for that particular task. However, performing cross task evaluation (i.e., suppressing cognitive heads for task like Retrieval and evaluating on Knowledge Recall) would provide more insights. Although a similar analysis in provided between lower and higher level cognitive functions, intra-level function relations would substantiate the Functional Clustering results presented in Section 4.3.**
>
> **A.** Following your suggestions, we conduct experiments where we mask the retrieval heads during the evaluation of knowledge recall (Recall), and conversely, mask knowledge recall heads during the evaluation of retrieval performance. As shown in the following table, intra-level masking of cognitive heads leads to a certain degree of performance degradation. However, as shown in Table 2, masking either Retrieval or Knowledge Recall heads results in a complete failure on high-level reasoning tasks, with performance dropping to zero.
>
> This stark contrast suggests that low-level cognitive heads not only play a crucial role within their own level but also significantly influence high-level functions through the chain-of-thought (CoT) reasoning process.
>
>
> **Table. Intervention results (%) of different cognitive heads and random heads across Retrieval and Recall functions.**
> |Model|Inter_Head|Retrieval (comet)|Retrieval (acc)|Recall (comet)|Recall (acc)|
> |-|-|-|-|-|-|
> |LLama3.1-8B|random|90.83|84.71|87.85|83.84|
> |LLama3.1-8B|retrieval|44.96|8.24|72.05|33.33|
> |LLama3.1-8B|recall|86.79|75.29|56.93|38.38|
> |Qwen3-8B|random|92.81|75.29|89.90|53.54|
> |Qwen3-8B|retrieval|59.19|38.24|79.26|57.58|
> |Qwen3-8B|recall|83.31|71.18|64.81|30.30|
>
>
> **Q7. Change in accuracy due to positive/negative interventions in attention heads corresponding to specific cognitive tasks would further solidify their analysis.**
>
> **A.** To further quantify the results, we conduct experiments on two tasks: a math task and a retrieval task. For the math task, we use 100 samples from GSM8K (denoted as GSM8K\_100), and for the retrieval task, we use an extractive QA dataset consisting of 49 samples, as described in [1]. The Extractive\_QA dataset is constructed by prompting GPT-4o to generate question–answer pairs from a given paragraph, where the answer is an original sentence extracted from that paragraph.
>
> For negative intervention, we suppress the Math Calculation heads when evaluating on GSM8K\_100, and the Retrieval heads when evaluating on Extractive\_QA. As shown in the following table, this suppression leads to a significant drop in performance on both tasks across different models, demonstrating that these heads exhibit consistent and transferable functional roles.
>
> **Table. LLM performance on GSM8k_100 and Extractive_QA by negative intervention. Inter. is the negative intervention.**
> |Dataset|Head|Llama3.1-8B|Llama3.2-3B|Qwen3-8B|Qwen3-4B|
> |-|-|-|-|-|-|
> |GSM8k\_100|Math (Base)|82|64|94|91|
> |GSM8k\_100|Math (Inter.)|38|34|34|37|
> |Extractive\_QA|Retrieval (Base)|57.14|36.73|57.14|51.02|
> |Extractive\_QA|Retrieval (Inter.)|0|0|14.29|12.24|
>
> For positive intervention, we intervene on the cognitive heads in LLMs using positive activation adjustments. The experimental results in the following table show that enhancing the activation of retrieval heads along their corresponding functional directions leads to improved performance on the retrieval task.
>
> For the math task, each question often involves multiple cognitive functions. We observe that in some cases, the numerical computations are correct, but the model’s understanding of the question is semantically flawed. By positively intervening on either the Math Calculation or Semantic Understanding heads, we observe consistent improvements in performance. This indicates the importance of developing intervention methods that can simultaneously adjust multiple cognitive functional heads which collaborate during complex tasks. We leave this as an avenue for future work.
>
> **Table. LLM performance on GSM8k_{100} and Extractive_QA by positive intervention. Inter. is the positive intervention.**
> |Dataset|Head|Llama3.1-8B|Llama3.2-3B|Qwen3-8B|Qwen3-4B|
> |-|-|-|-|-|-|
> |GSM8K|Math(Base)|82|64|91|94|
> |GSM8K|Math(Inter.)|84|66|92|94|
> |GSM8K|Semantic(Base)|82|64|91|94|
> |GSM8K|Semantic(Inter.)|84|65|93|94|
> |Extractive_QA|Retrieval(Base)|57.14|36.73|57.14|51.02|
> |Extractive_QA|Retrieval(Inter.)|63.26|44.90|61.22|69.38|
>
> [1] Wu, Wenhao, et al. "Retrieval head mechanistically explains long-context factuality." arXiv preprint arXiv:2404.15574 (2024).

---

> ### Author Response · Authors · 2025-08-04
> **Response to Reviewer 8YmT**
>
> Dear Reviewer,
>
> Thank you very much for taking the time and effort to review our paper. In response to your suggestions, we will incorporate additional baselines and quantitative experiments on both positive and negative interventions to further enhance our analysis in the final version.
>
> As the discussion deadline approaches, we would be grateful for any further comments or suggestions you may have for improving our work. Your feedback is highly valuable, and we sincerely welcome your continued guidance.
>
> Thank you once again for your thoughtful review and support.
>
> Best regards,
>
> The Authors

---

> > ### Comment · Reviewer_8YmT · 2025-08-05
> >
> > Thanks you for detailed explanations. I have a few more questions:
> >
> > 1. Regarding Q7: Can the authors comment on the relation between the size of the model and the difference in performances? It looks like the difference between the intervened model and base model is high if the parameters are more, irrespective of the type of intervention.
> >
> > 2. Regarding knowledge recall: On what basis do the authors mention that the cases are rare that the knowledge is not present in the model? This evaluation can be faulty on two lines: a) The relevant knowledge is not present in the model b) Even if the knowledge is present, the model is not able to retrieve (maybe because of less frequency during training). Do the results account for this phenomenon and only pick questions that are always answered by the model?

---

> ### Author Response · Authors · 2025-08-05
> **Response to Reviewer 8YmT**
>
> Dear Reviewer,
>
> Thank you for your thoughtful follow-up and for engaging with our detailed responses. We sincerely appreciate your feedback and would like to provide additional clarification on the points you raised.
>
> **1. Can the authors comment on the relation between the size of the model and the difference in performances? It looks like the difference between the intervened model and base model is high if the parameters are more, irrespective of the type of intervention.**
>
> We observe that after masking the cognitive heads, the performance of the intervened models drops to a similarly low level across different LLMs, regardless of the original model size or base performance. This is expected, as the crucial cognitive heads responsible for specific functions are disabled, making it difficult for the model to arrive at correct answers.
>
> For the math task, the remaining ~30\% accuracy after intervention likely arises from two factors: (1) the base model may have memorized certain answers, and (2) some simpler questions do not require math calculation.
>
> For the retrieval task, once the Retrieval heads are masked, the model's retrieval ability is almost entirely lost across all model sizes. This indicates that cognitive functions are indeed localized in a subset of heads, and masking them leads to a systematic degradation, irrespective of model capacity.
>
> **2. Regarding knowledge recall: On what basis do the authors mention that the cases are rare that the knowledge is not present in the model? This evaluation can be faulty on two lines: a) The relevant knowledge is not present in the model b) Even if the knowledge is present, the model is not able to retrieve (maybe because of less frequency during training). Do the results account for this phenomenon and only pick questions that are always answered by the model?**
>
> Thank you for your thoughtful question. To clarify, we computed the accuracy for the Knowledge Recall task and observed relatively high scores: 85.4\% for LLaMA3.1-8B, 75.2\% for LLaMA3.2-3B, 89.1\% for Qwen-8B, and 86.3\% for Qwen-4B. Given the high accuracy and, as noted in our response to Q4, that "even if the model does not know the answer, it still engages in the process of attempting to recall factual knowledge—much like a human trying to retrieve uncertain information," we did not explicitly filter out failure cases. Our experimental results in the paper demonstrate the effectiveness of our method: when masking the identified Knowledge Recall heads, model performance on relevant tasks drops significantly.
>
> While we cannot definitively determine whether failures in Recall are due to the absence of knowledge in the model or from recall issues, as also noted in Q4, "incorrect answers may correspond to insignificant cognitive head activation, so the straightforward approach is to exclude them from the analysis.".
>
> Thank you once again for your thoughtful review and consideration.
>
> Best regards,
>
> The Authors

---

> > ### Comment · Reviewer_8YmT · 2025-08-06
> >
> > Thank you for the responses.
> >
> > I kindly request the authors to include the discussion on cognitive functions and some analysis on size of the model vs interventions in a revised version of the paper. I would like to maintain my score.

---

> > > ### Author Response · Authors · 2025-08-06
> > >
> > > Dear Reviewer,
> > >
> > > We will certainly include the discussion on cognitive functions and the analysis of model size versus intervention effects in the revised version of the paper. Considering that all comments have been addressed, we kindly expect you might to reconsider the rating. Thanks again for your constructive feedback and your time reviewing our work.
> > >
> > > Best Regards,
> > >
> > > The Authors

---

### Official Review · Reviewer_Nip4 · 2025-06-30

**Clarity:** 3
**Significance:** 3
**Originality:** 3
**Rating:** 4
**Confidence:** 4

**Summary:**

This paper introduces a new dataset for evaluating the cognitive roles of different attention heads in large language models (LLMs). In particular, the dataset is constructed by prompting ChatGPT to decompose original questions into sub-questions focusing on different roles (e.g., retrieval and logical reasoning). Its analysis centers around training linear classifiers to predict the cognitive roles based on features from different heads and layers, and estimating the importance of attention heads with gradient-based saliency methods. Experiments highlight several interesting characteristics of the attention heads, e.g., sparsity of important heads and their structure across layers.

**Questions:**

(1) Please clarify the usage of subquestion and linear classifier during the analysis.

(2) What are most of the important attention heads located in a few middle layers regardless of cognitive roles?

(3) Please consider performing the masking experiments with a different baseline (as suggested in the comments above).

(4) It would be reasonable to include more quantitative analysis for the positive/negative interventions.

**Ethical Concerns:**

["NO or VERY MINOR ethics concerns only"]

**Final Justification:**

The rebuttal addresses most of my concerns raised in the initial review, and I have decided to raise my score.

**Limitations:**

yes

**Quality:**

3

**Strengths And Weaknesses:**

As far as I am concerned, this paper has the following strengths:

(1) It is an intuitive idea to understand LLMs based on the distinct roles of their attention heads. The paper performs a collection of analyses, and facilitates understanding from different perspectives.

(2) The paper presents a new dataset, which could be useful for subsequent studies of related fields.

(3)  It is interesting that manipulating attention heads with lower-level cognitive roles leads to considerable impacts on higher-level cognitive reasoning.

However, it also has several limitations that can be improved:

(1) It is not very clear how the hierarchical structure experiment is performed (Table 2). Are the authors consistently manipulating attention heads of lower cognitive functions across all subquestions, or only permuting the initial subquestions related to low-level cognitive roles and evaluating decision-making questions after? If it is the former, how would attention heads affect the chain-of-thought reasoning?

(2) This may be related to the previous comment. What are the key advantages of question decomposition? It is also feasible to determine the cognitive roles of the original questions without decomposition.

(3) The authors argue that the attention heads have layered functional organization. Nevertheless, this is mostly obvious only for math vs other cognitive functions. Looking at Figure 2 and Figure 5-9, it appears that the important attention heads are commonly located within a few layers (e.g., layer 13-15 in Figure 2), regardless of cognitive functions and models. Any idea why these layers are so special?

(4) Table 1 shows that masking important attention heads leads to significant performance drop, which random masking does not. However, this could be related to the observations that important attention heads are concentrated in a few layers, without considering the cognitive roles (e.g., randomly masking attention heads in layer 1-10 may not affect the performance, while masking layers 13-15 would). I would suggest considering a different baseline, which performs the permutation based on attention heads identified by a different cognitive role (e.g., masking attention heads for retrieval while evaluating the performance for recall).

(5) Analysis in Section 4.4 looks interesting, but only two qualitative examples are provided. In addition, both of them are about math calculation.

(6) The proposed method simultaneously considers features from all attention heads across layers during linear probing, which leads to inputs of very high dimension. Considering the size of the dataset (~3000 samples), do the linear classifiers have overfitting issues? What is the accuracy of these classifiers on predicting the cognitive roles?

---

> ### Author Rebuttal · Authors · 2025-07-31
>
> Dear Reviewer,
>
> We deeply appreciate the time and effort you have dedicated to reviewing our work. Below, we address the questions you raised.
>
> **Q1.  It is not very clear how the hierarchical structure experiment is performed (Table 2). Are the authors consistently manipulating attention heads of lower cognitive functions across all subquestions, or only permuting the initial subquestions related to low-level cognitive roles and evaluating decision-making questions after? If it is the former, how would attention heads affect the chain-of-thought reasoning?**
>
> **A.** For the hierarchical structure experiment—such as analyzing the influence of Retrieval heads on Mathematical Calculation—we intervene by masking Retrieval-related heads across all subquestions in the LLM. The answers generated from earlier subquestions then serve as prior information for subsequent subquestions involving math calculation. This setup enables us to investigate how suppressing low-level cognitive heads (e.g., Retrieval) can indirectly impact higher-level functions (e.g., Math Calculation) by propagating through the chain of reasoning.
>
> Additionally, as shown in Table 1, randomly masking certain heads can also have a direct effect on cognitive functions.
>
> Thus, in this experiment, low-level cognitive heads influence high-level reasoning via two mechanisms: (1) indirectly, by affecting intermediate subquestion answers within the chain-of-thought process; and (2) directly, by masking a part of heads.
>
> **Q2. This may be related to the previous comment. What are the key advantages of question decomposition? It is also feasible to determine the cognitive roles of the original questions without decomposition.  Please clarify the usage of subquestion and linear classifier during the analysis.**
>
> **A.** Thank you for the valuable question. We would like to clarify the key advantages of using question decomposition in our analysis:
>
> 1. Reasoning tasks typically involve the presence—but rarely the explicit absence—of certain cognitive functions. Treating the full question as a single unit results in a lack of well-defined negative samples (i.e., tasks that do not require a given function), which makes probing-based analysis less effective. Moreover, incorrect answers do not necessarily serve as reliable negative examples, since a wrong answer can arise from various causes unrelated to the absence of a cognitive function. By decomposing questions into subquestions, we obtain fine-grained annotations that clearly delineate the cognitive functions involved in each part, enabling precise classification and more effective probing.
>
> 2. Reasoning tasks are diverse, with different tasks involving different and often multiple cognitive functions. Subquestions allow us to disentangle these functions and examine them systematically and independently, thereby reflecting a more human-like interpretable model.
>
> 3. Importantly, question decomposition facilitates building a comprehensive interpretability framework that accounts for both the independent and collaborative roles of attention heads across cognitive functions. This framework offers practical benefits for model design, modular training, and fine-tuning strategies.
>
> **Q3. The authors argue that the attention heads have layered functional organization. Nevertheless, this is mostly obvious only for math vs other cognitive functions. Looking at Figure 2 and Figure 5-9, it appears that the important attention heads are commonly located within a few layers (e.g., layer 13-15 in Figure 2), regardless of cognitive functions and models. Any idea why these layers are so special? What are most of the important attention heads located in a few middle layers regardless of cognitive roles?**
>
> From Figures 2 and 5–9, we observe that many cognitively important attention heads are located in the middle layers. This finding is consistent with [1], which suggests that intermediate layers can encode even richer representations. A recent study on VLMs [2] further supports this, demonstrating that middle layers play a critical role. Notably, while math-related heads tend to emerge at deeper layers, other high-level cognitive functions—such as decision-making—also become more prominent in the higher layers for some LLMs. In general, shallow layers primarily capture surface-level patterns, whereas mid-to-late layers are responsible for constructing task-specific and abstract representations closely tied to cognitive processing.
>
> [1] Skean, Oscar, et al. "Layer by layer: Uncovering hidden representations in language models." ICML 2025.
>
> [2] Jiang, Zhangqi, et al. "Devils in middle layers of large vision-language models: Interpreting, detecting and mitigating object hallucinations via attention lens." CVPR 2025.
>
> **Q4.  Please consider performing the masking experiments with a different baseline.**
>
> **A.** To further validate the functional roles of cognitive heads, we conduct experiments where we mask the retrieval heads during the evaluation of knowledge recall (Recall), and conversely, mask knowledge recall heads during the evaluation of retrieval performance. As shown in the following table, masking the corresponding cognitive heads leads to a significantly larger performance drop compared to masking other heads, supporting their functional specialization.
>
>
> **Table. Intervention results (%) of different cognitive heads and random heads across Retrieval and Recall functions.**
> |Model|Inter_Head|Retrieval (comet)|Retrieval (acc)|Recall (comet)|Recall (acc)|
> |-|-|-|-|-|-|
> |LLama3.1-8B|random|90.83|84.71|87.85|83.84|
> |LLama3.1-8B|retrieval|44.96|8.24|72.05|33.33|
> |LLama3.1-8B|recall|86.79|75.29|56.93|38.38|
> |Qwen3-8B|random|92.81|75.29|89.90|53.54|
> |Qwen3-8B|retrieval|59.19|38.24|79.26|57.58|
> |Qwen3-8B|recall|83.31|71.18|64.81|30.30|
>
> **Q5. Analysis in Section 4.4 looks interesting, but only two qualitative examples are provided. In addition, both of them are about math calculation.**
>
> **A.** To further quantify the results, we conduct experiments on two tasks: a math task and a retrieval task. For the math task, we use 100 samples from GSM8K (denoted as GSM8K\_100), and for the retrieval task, we use an extractive QA dataset consisting of 49 samples, as described in [1]. The Extractive\_QA dataset is constructed by prompting GPT-4o to generate question–answer pairs from a given paragraph, where the answer is an original sentence extracted from that paragraph.
>
> For negative intervention, we suppress the Math Calculation heads when evaluating on GSM8K\_100, and the Retrieval heads when evaluating on Extractive\_QA. As shown in the following table, this suppression leads to a significant drop in performance on both tasks across different models, demonstrating that these heads exhibit consistent and transferable functional roles.
>
> **Table. LLM performance on GSM8k_100 and Extractive_QA by negative intervention. Inter. is the negative intervention.**
> |Dataset|Head|Llama3.1-8B|Llama3.2-3B|Qwen3-8B|Qwen3-4B|
> |-|-|-|-|-|-|
> |GSM8k\_100|Math (Base)|82|64|94|91|
> |GSM8k\_100|Math (Inter.)|38|34|34|37|
> |Extractive\_QA|Retrieval (Base)|57.14|36.73|57.14|51.02|
> |Extractive\_QA|Retrieval (Inter.)|0|0|14.29|12.24|
>
> For positive intervention, we intervene on the cognitive heads in LLMs using positive activation adjustments. The experimental results in the following table show that enhancing the activation of retrieval heads along their corresponding functional directions leads to improved performance on the retrieval task.
>
> For the math task, each question often involves multiple cognitive functions. We observe that in some cases, the numerical computations are correct, but the model’s understanding of the question is semantically flawed. By positively intervening on either the Math Calculation or Semantic Understanding heads, we observe consistent improvements in performance. This indicates the importance of developing intervention methods that can simultaneously adjust multiple cognitive functional heads which collaborate during complex tasks. We leave this as an avenue for future work.
>
> **Table. LLM performance on GSM8k_100 and Extractive_QA by positive intervention. Inter. is the positive intervention.**
> |Dataset|Head|Llama3.1-8B|Llama3.2-3B|Qwen3-8B|Qwen3-4B|
> |-|-|-|-|-|-|
> |GSM8K|Math(Base)|82|64|91|94|
> |GSM8K|Math(Inter.)|84|66|92|94|
> |GSM8K|Semantic(Base)|82|64|91|94|
> |GSM8K|Semantic(Inter.)|84|65|93|94|
> |Extractive_QA|Retrieval(Base)|57.14|36.73|57.14|51.02|
> |Extractive_QA|Retrieval(Inter.)|63.26|44.90|61.22|69.38|
>
> [1] Wu, Wenhao, et al. "Retrieval head mechanistically explains long-context factuality." arXiv preprint arXiv:2404.15574 (2024).
>
> **Q6. The proposed method simultaneously considers features from all attention heads across layers during linear probing, which leads to inputs of very high dimension. Considering the size of the dataset (~3000 samples), do the linear classifiers have overfitting issues? What is the accuracy of these classifiers on predicting the cognitive roles?**
>
> Our experimental results show that the average accuracy for predicting cognitive roles on test dataset of CogQA is around 80\% which demonstrates that the classifiers can effectively capture the relevant features.
>
> **Table. Test accuracy for LLMs.**
> | Dataset | Llama3.2-3B | Qwen3-8B | Yi-1.5-9B |
> |---------|-------------|-------------|----------|
> | CogQA   | 79.80       | 84.71    | 80.79    | 77.56     |

---

> > ### Comment · Reviewer_Nip4 · 2025-08-05
> >
> > I thank the authors for the clarification and additional experiments. The rebuttal resolves most of my concerns in the initial reviews, e.g., more in-depth analysis and linear probing. I do have a followed-up questions:
> >
> > Looking at the results for the new masking experiments, it appears that masking based on another function lead to significantly larger effects than random masking. To me, this is also related to my initial question regarding the ambiguous layer organization, where many functions shared similar focuses. I wonder if the authors have some comments about disentangling the attention head/focuses in the middle-layer.

---

> > > ### Author Response · Authors · 2025-08-06
> > > **Response to Reviewer Nip4**
> > >
> > > Dear Reviewer,
> > >
> > > Thank you for your follow-up and for acknowledging our clarifications and additional experiments.
> > >
> > > **Regarding your observation on new masking experiments, and if the authors have some comments about disentangling the attention head/focuses in the middle-layer.**
> > >
> > > Thank you for your thoughtful observation. Indeed, your point—that masking attention heads associated with another function leads to significantly greater performance drops than random masking—aligns with our findings in the Functional Clustering analysis (Subsection 4.3). This suggests that certain cognitive heads—such as those associated with Retrieval—can be more influential to other tasks like Knowledge Recall than randomly selected, as both functions are low-level and serve complementary roles in information extraction and understanding. The differences observed between LLaMA3.1-8B and Qwen-8B further indicate that the degree of such influence may vary across model architectures.
> > >
> > > From Figure 2 and Figures 5–9 in the Appendix, we observe that many cognitive heads exhibit a degree of functional disentanglement, as different functions tend to activate distinct sets of heads (yellow highlighted part). The results of our intervention experiments also support this: masking heads specific to one function yields targeted performance degradation, which would be unlikely if heads were highly entangled.
> > >
> > > We also note that a certain degree of entanglement between cognitive functions—particularly in the middle layers—is biologically plausible and consistent with how the human brain organizes related processes.
> > >
> > > Looking ahead, we believe further analysis of these identified heads at a coarser granularity could be insightful. For instance, identifying whether some shared heads are responsible for global control mechanisms (analogous to executive function in the brain) or for language structuring and output could open new directions for understanding modularity in LLMs.
> > >
> > > We hope this response addresses your question. If there are any remaining concerns, we would be happy to continue the discussion. Looking forward to your response and changes.
> > >
> > > Best regards
> > >
> > > The Authors

---

> ### Author Response · Authors · 2025-08-04
> **Response to Reviewer Nip4**
>
> Dear Reviewer,
>
> Thank you for taking the time and effort to review our paper.
> Following your suggestions, we will include extra baselines and quantitative experiments on both positive and negative interventions to further strengthen our analysis in the final version.
>
> As the discussion deadline approaches, we would greatly appreciate any further suggestions you might have for improving our work. Your feedback is invaluable, and we warmly welcome your guidance.
>
> Thank you once again for your thoughtful review and consideration.
>
> Best regards,
>
> The Authors

---

### Official Review · Reviewer_wHF7 · 2025-06-30

**Clarity:** 3
**Significance:** 3
**Originality:** 3
**Rating:** 5
**Confidence:** 4

**Summary:**

The paper investigates the cognitive role of attention heads in Transformers. The central assumption is that transformers exhibit cognitive functions analogous to those in humans, and that these functions are embedded within attention heads. To explore this hypothesis, the authors design a dataset where each question is annotated with a specific cognitive function. They then use a linear probe classifier to identify the cognitive role of each attention head. In the experiments, the authors analyze the distribution of cognitive functions across attention heads and perform interventions to evaluate the impact of individual heads on model performance.

**Questions:**

Please see the weaknesses part.

**Ethical Concerns:**

["NO or VERY MINOR ethics concerns only"]

**Final Justification:**

The authors provide sufficient evidence and details. I have no other concerns.

**Limitations:**

Yes, the authors include a discussion of the limitations of their work.

**Quality:**

3

**Strengths And Weaknesses:**

**Strengths**
- The paper connects insights from cognitive science with interpretability research, which is an interesting direction.
- The idea of linking attention heads to cognitive functions is intriguing.
- The paper introduces a new dataset, which may be valuable for future research.

**Weaknesses**
- The definitions of each cognitive function are vague and ambiguous. The proposed categories may not be atomic or mutually exclusive. For example, decision-making and inference often co-occur with other classes, and it is unclear why mathematical computation would be considered distinct from or incompatible with logical reasoning. Applying a classifier to force these functions into exclusive categories may not be a valid approach.

- The method of using GPT-4o-mini to "extract key words" is not a sound approach:
  - It is unclear whether key words alone can fully capture or identify a cognitive function.
  - Even if key words do provide a comprehensive representation in some cases, it is difficult to justify that GPT-4o-mini and the target model share the same mechanism or metric for determining word importance.

- It is unclear why adding the layer-averaged attention vector as a feature would resolve the issue of cognitive functions varying by depth, given that this information still originates from the same layer. Moreover, since the goal is to identify the distinct role of each attention head, incorporating the average activation may fail to disentangle the influence of individual heads from the collective effect of the layer.

- The underlying assumption that each attention head corresponds to a single cognitive function is quite strong. A counterexample is the first layer of an induction head, whose role is primarily to provide contextual information rather than perform a distinct cognitive function.

- It is unclear how the reported accuracy is calculated. If accuracy is based solely on whether the model answers correctly, it becomes difficult to attribute failure cases to the cognitive function roles of specific attention heads. Suppressing an essential head across all examples could lead to uniformly degraded performance, as shown in Table 1, without offering insight into functional specialization.

- The paper derives its results from a single dataset. To demonstrate that the identified functional roles are generalizable, the authors should evaluate the attention heads—identified using the proposed dataset—on out-of-distribution tasks. For example, they could suppress the attention head associated with mathematical calculation and assess its impact on a benchmark like GSM8K to test whether the identified function holds in a different setting.

- Regarding the steering experiment in Section 4.4, quantitative results evaluated on multiple examples are needed, rather than relying on a single-case study.


- Reproducibility is limited, as the authors do not provide code at the time of review.

**Minor:**

- In line 138, the set annotation should be under the summation symbol.

---

> ### Author Rebuttal · Authors · 2025-07-31
>
> Dear Reviewer,
>
> We sincerely appreciate the time and effort you’ve invested in reviewing our paper. Below, we provide detailed responses to the questions you raised.
>
> **Q1. About definitions of Cognitive Functions and the Validity of the proposed method.**
>
> **A.** 1. Inspired by cognitive science [1-2] and recent LLM attention research [3], we focus on eight key cognitive functions, split into low- and high-level categories (see Subsection 2.1). To better align with human cognitive processes, we incorporate human annotations to guide the function labeling for subquestions. A brief overview of these functions is provided in subsection 2.1, with details in Appendix A.4.
>
> 2. About decision-making: we define it as "the process of making a choice in a selection task based on prior information or analysis." This prior information often results from other cognitive processes such as inference or mathematical computation. Therefore, in decision-making tasks (e.g., multiple-choice questions), the model typically does not need to perform new reasoning, but rather selects the correct answer based on the output of earlier processes. This aligns with our observation that decision-making is frequently associated with retrieval and inference/logical reasoning, consistent with its cognitive definition (see Subsection 4.3). Importantly, cognitive functions in LLMs—like those in the human brain—are inherently complex and interdependent.
>
> Furthermore, we distinguish between mathematical computation and logical reasoning. For instance, the question “What is the total amount Lee earned from mowing 16 lawns?” requires mathematical computation, whereas “Can a man who is swimming also go hiking?” involves logical reasoning. These tasks have distinct cognitive demands and should not be conflated.
>
> 3. In our current setting, we treat each instance as primarily reflecting a dominant cognitive function to facilitate analysis and probing, rather than to assert hard functional boundaries. Our experimental results in subsection 4.2 and 4.4 in the paper demonstrate the effectiveness of this approach. An additional experiment investigating the influence of masking one class of cognitive heads on other cognitive tasks (Response to Q5) further demonstrates that the cognitive heads are indeed relatively disentangled.
>
> **Q2. About using GPT-4o-mini to extract key words.**
>
> **A.** 1. In CogQA, each subquestion is annotated with a predefined cognitive function, which is determined primarily based on the subquestion itself. Furthermore, certain attention heads are more likely to be activated when generating key tokens associated with specific cognitive functions, thus, we use GPT-4o-mini to extract key tokens from the generated subanswers, to help identify which attention heads are activated during the generation of these function-relevant tokens.
>
> 2. We do not assume that GPT-4o-mini and the target model share identical importance metrics. Rather, we use these tokens to isolate content-relevant attention patterns and to reduce noise in the probing process.
>
> **Q3. About adding the layer-averaged attention vector.**
>
> **A.** It has been suggested that cognitive functions may vary by layer depth [3], meaning that even if two attention heads across different layers exhibit similar activation patterns aligned with a cognitive function, they might still play distinct roles due to their positional context in the model. Thus, layer information is crucial for interpreting functional specialization. We include the layer vector as a coarse signal to differentiate these roles.
>
> We compare using the layer-averaged vector (Layer_True) versus not (Layer_False) in the following table. Masking heads with Layer_True causes a larger accuracy drop, showing the value of including layer information.
>
> **Table. The results of Llama3.1-8B-instruct when masking Retrieval heads.**
> |Model|Inter_Head|comet|acc|
> |-|-|-|-|
> |Llama3.1-8B|layer_True|70.05|46.47|
> |Llama3.1-8B|layer_False|69.88|53.53|
>
> **Q4. About the underlying assumption of attention head and the example of induction head**
>
> **A.** For the example of induction head, while certain heads in the first layer are primarily responsible for providing contextual information that is also related to induction, there exist some heads that are directly involved in performing induction. While they are both related to induction, the difference in their roles reflects their varying importance for the induction task. In other words, masking an induction head with high functional importance tends to result in a greater performance drop than masking a lower-layer contextual head.
>
> In our approach, we quantify and rank the importance of attention heads for a given cognitive function, identifying the top ones as cognitive heads. This does not imply that all other heads are entirely non-functional or irrelevant; rather, they contribute less significantly compared to the top-ranked ones.
>
> Importantly, while prior work often focuses on a single function or mechanism, our study extends to multiple cognitive functions and adopts a more fine-grained perspective. We aim to reveal dominant functional tendencies in heads, not strict one-to-one mappings.  We will clarify and illustrate this in the revised version.
>
> **Q5.  About accuracy and evidence for functional specialization.**
>
> **A.** As described in Subsection 4.2 (lines 202–206), we use several standard evaluation metrics, including COMET, BLEU, ROUGE, and semantic similarity (measured by the cosine similarity of embeddings). The overall accuracy is computed as a combination of BLEU, ROUGE, and semantic similarity scores.
>
> Beyond Table 1, examples in Subsection 4.4 demonstrate that a positive intervention on Math Calculation heads in Qwen3-4B can correct originally incorrect mathematical computations, whereas a negative intervention can cause originally correct computations to become incorrect. Notably, the examples used in Subsection 4.4 are not from the CogQA dataset.
>
> To further validate the functional roles of cognitive heads, we conduct experiments where we mask the retrieval heads during the evaluation of knowledge recall (Recall), and conversely, mask knowledge recall heads during the evaluation of retrieval performance. As shown in the following table, masking the corresponding cognitive heads causes a significantly larger performance drop than masking others, confirming their functional specialization.
>
> **Table. Intervention results (%) of different cognitive heads and random heads across Retrieval and Recall functions.**
> |Model|Inter_Head|Retrieval (comet)|Retrieval (acc)|Recall (comet)|Recall (acc)|
> |-|-|-|-|-|-|
> |LLama3.1-8B|random|90.83|84.71|87.85|83.84|
> |LLama3.1-8B|retrieval|44.96|8.24|72.05|33.33|
> |LLama3.1-8B|recall|86.79|75.29|56.93|38.38|
> |Qwen3-8B|random|92.81|75.29|89.90|53.54|
> |Qwen3-8B|retrieval|59.19|38.24|79.26|57.58|
> |Qwen3-8B|recall|83.31|71.18|64.81|30.30|
>
> **Q6. Experiments to demonstrate that the identified functional roles are generalizable.**
>
> **A.** Thank you for your suggestions. To evaluate the generalization of the identified functional heads, we present results on data beyond the CogQA dataset in Subsection 4.4. To further quantify the results, we conduct experiments on two tasks: a math task using 100 GSM8K samples (GSM8K_100) and a retrieval task with 49 samples from an extractive QA dataset. The Extractive_QA pairs are generated by GPT-4o, with answers extracted directly from the source paragraph.
>
> We suppress Math Calculation heads for GSM8K_100 and Retrieval heads for Extractive_QA. As shown in the table, this causes significant performance drops across models, confirming these heads’ consistent functional roles. Math accuracy is measured by exact answer match; Extractive_QA accuracy checks if the original paragraph answer appears in the response.
>
> **Table. LLM performance on GSM8k_100 and Extractive_QA by negative intervention. Inter. is the negative intervention.**
> |Dataset|Head|Llama3.1-8B|Llama3.2-3B|Qwen3-8B|Qwen3-4B|
> |-|-|-|-|-|-|
> |GSM8k\_100|Math (Base)|82|64|94|91|
> |GSM8k\_100|Math (Inter.)|38|34|34|37|
> |Extractive\_QA|Retrieval (Base)|57.14|36.73|57.14|51.02|
> |Extractive\_QA|Retrieval (Inter.)|0|0|14.29|12.24|
>
> **Q7. More quantitative results about steering experiment.**
>
> **A.** We intervene on the cognitive heads in LLMs using activation adjustments on the GSM8K_100 and Extractive_QA datasets. The table below shows that boosting retrieval head activations along their functional directions improves retrieval task performance.
>
> Math tasks often involve multiple cognitive functions. We observe that in some cases, the numerical computations are correct, but the model’s understanding of the question is semantically flawed. By positively intervening on either the Math Calculation or Semantic Understanding heads, we observe consistent improvements in performance. This highlights the need for intervention methods that adjust multiple cognitive heads simultaneously. We leave this for future work.
>
> **Table. LLM performance on GSM8k_{100} and Extractive_QA by positive intervention. Inter. is the positive intervention.**
> |Dataset|Head|Llama3.1-8B|Llama3.2-3B|Qwen3-8B|Qwen3-4B|
> |-|-|-|-|-|-|
> |GSM8K|Math(Base)|82|64|91|94|
> |GSM8K|Math(Inter.)|84|66|92|94|
> |GSM8K|Semantic(Base)|82|64|91|94|
> |GSM8K|Semantic(Inter.)|84|65|93|94|
> |Extractive_QA|Retrieval(Base)|57.14|36.73|57.14|51.02|
> |Extractive_QA|Retrieval(Inter.)|63.26|44.90|61.22|69.38|
>
> **Q8. About code and the set annotation in line 138.**
>
> We will open-source our code upon acceptance to ensure reproducibility. Thanks for noting the mistake in line 138; we will correct it and review the manuscript thoroughly.
>
> [1] John R Anderson. Rules of the mind. Psychology Press, 2014.
>
> [2] Adele Diamond. Executive functions. Annual review of psychology, 64(1):135–168, 2013
>
> [3] Zheng, Z., et al. "Attention heads of large language models: A survey. arXiv 2024." arXiv preprint arXiv:2409.03752.

---

> > ### Author Response · Authors · 2025-08-04
> > **Response to Reviewer wHF7**
> >
> > Dear Reviewer,
> >
> > Thank you very much for your time and effort in reviewing our paper. We have carefully addressed all the questions you raised and hope our responses have addressed your concerns.
> >
> > As the discussion deadline approaches, we kindly inquire if you have any additional questions or suggestions for improving our manuscript. Your input is highly valued, and we would greatly appreciate your guidance.
> >
> > Thank you again for your thoughtful review and consideration.
> >
> > Best regards,
> >
> > The Authors

---

> > ### Comment · Reviewer_wHF7 · 2025-08-05
> >
> > I thank the authors for the detailed responses and the hard work. In detail,
> >
> > **Regarding A1:** The definitions of cognitive abilities are well accepted, as is the hierarchy structure. I still have a few doubts regarding the claims:
> > - If cognitive functions' definitions are hierarchical in nature, why would they be distinct classes when applying the classifier?
> > - *"Cognitive functions in LLMs—like those in the human brain—are inherently complex and interdependent."*: Given the polysementicity nature of the neurons, the chance of an attention head serving multiple roles, I think, is quite high. In this case, how does the current framework address this behavior?
> > - From another perspective, I wonder if there is a possibility that an attention head server has no role in the defined cognitive abilities, given that they are most likely not a complete set of cognition. The framework seems not to allow this case but forces them into the pre-defined categories. It would be more assuring if the "void" case is allowed.
> >
> > **Regarding A2:** I am not fully convinced by the responses. To clarify, I am concerned
> > - The misalignment between the GPT-o4-mini model and the evaluated models. As they are different systems, a feature that is important to the GPT-o4-mini model may not be guaranteed to be the one for the evaluated models.
> > - How good are the tokens to represent defined cognitive functions? I would appreciate a case study that shows the extracted tokens for each cognitive function category.
> >
> > **Regarding A3, A4, A5, A6, A7** Thank you. I have no other questions regarding this part.
> >
> > **Regarding Q8:** Thank you. But if the replication details are missing upon reviewing, I have to maintain my evaluation.

---

> ### Author Response · Authors · 2025-08-05
> **Response to Reviewer wHF7**
>
> Dear Reviewer,
>
> Thank you very much for your thoughtful follow-up and for acknowledging our detailed responses. We appreciate your comments and would like to offer further clarification on the points raised.
>
> **Regarding A1**
>
> **1. If cognitive functions' definitions are hierarchical in nature, why would they be distinct classes when applying the classifier?**
>
> We would like to clarify this with the example of the math task shown in Fig. 1 of our paper. Solving this type of question requires a combination of cognitive functions, including low-level functions: Knowledge Recall (retrieving relevant mathematical facts) and Semantic Understanding (interpreting the problem statement), and high-level functions: Math Calculation, and ultimately Decision-Making (make a choice).
>
> A LLM may fail to answer correctly due to deficiencies in any of these components. As we discussed in our response to Q7, errors in GSM8K math questions can stem from different cognitive failures. Through targeted positive interventions using the corresponding cognitive heads, we can correct the model's output.
>
> Importantly, such classification allows us to better understand LLM behavior, and it provides a foundation for research, including reducing hallucinations (e.g., balancing long-term knowledge in Knowledge Recall with short-term Retrieval) and designing better COT prompting strategies.
>
> **2. "Given the polysementicity nature of the neurons, the chance of an attention head serving multiple roles, I think, is quite high. In this case, how does the current framework address this behavior?**
>
> In our approach, we quantify and rank the importance of attention heads for a given cognitive function. Heads with the highest importance scores are identified as cognitive heads for that function. Thus, an attention head that ranks highly for one cognitive function may also exhibit non-negligible importance for others, our framework allows this.
>
> **3. if the "void" case is allowed.**
>
> Our framework actually consider this "void" case.
> From Fig. 2 and Fig. 5-9 in Appendix, we can see that there are a lot of heads in the shallow layers that are "void" for any function due to their low importance to any class.
>
> **Regarding A2**
>
> **1. The misalignment between GPT-o4-mini and the evaluated models.**
>
> This may be a misunderstanding. As presented in lines 128-139 in Subsection 3.1, all activation features used in our framework are extracted directly from the evaluated models—not from GPT-4-o-mini. For each answer (which contains multiple generated tokens), we collect the corresponding activation features of attention heads from the evaluated model.
>
> The role of GPT-4-o-mini is solely to identify the most important tokens in the generated answers. We then use the activation features of those selected tokens (from the evaluated model) for classification. Generally, existing work [1] uses the activation features of the attention heads corresponding to the first generated token.
>
> [1] Li, Kenneth, et al. "Inference-time intervention: Eliciting truthful answers from a language model." NeurIPS 2023.
>
> **2.  Case study that shows the extracted tokens for each cognitive function.**
>
> The selected tokens are intended to semantically represent the generated answer. Below are examples for different cognitive functions for Llama3.1-8B-instruct:
>
> *main question: "Given the sentence \"A surfboarder catches the waves.\" can we conclude that \"A surfboarder in the water.\"?\\nOptions:\\n- yes\\n- it is not possible to tell\\n- no"*
>
> subquestion: "What is typically required for a surfboarder to catch waves?"
>
> cognitive function: "Knowledge Recall"
>
> answer: "The surfboarder needs to be in the water."
>
> selected tokens: ['surfboarder', 'needs', 'be', 'in', 'water']
>
> *main question: "Is the following a factual statement?\\n\"Due to its high density, countries around the world use Palladium to mint coins.\"\\nOptions:\\n- yes\\n- no"*
>
> subquestion: "What is the statement in question?"
>
> answer: "The statement in question is: Due to its high density, countries around the world use Palladium to mint coins."
>
> cognitive function: "Retrieval"
>
> selected tokens: ['high', 'density', 'Palladium', 'mint', 'coins']
>
> *main question: "A one-year subscription to a newspaper is offered with a 45\% discount. How much does the discounted subscription cost if a subscription normally costs \$80?"*
>
> subquestion: "How much is the discount amount in dollars for the subscription?"
>
> answer: "36"
>
> cognitive function: "Math Calculation"
>
> selected tokens: ['36']
>
> We can see that the selected tokens semantically represent the answer. Note that we use all tokens when the number of tokens is fewer than 5. We will add these examples in Appendix in the final version.
>
> Regarding Q8: We respect the reviewer's concerns. Please allow us to clarify that we indicated in the Checklist: "we will release the complete GitHub repository with all code upon paper acceptance."
>
> Thanks for your time.
>
> Best Regards,
>
> The Authors

---

> > ### Comment · Reviewer_wHF7 · 2025-08-06
> >
> > - **Regarding A1:** Thanks for the response. I have no other questions regarding this part.
> > - **Regarding A2:** Thanks for the details. My concern is addressed. And I would agree that adding the examples would further strengthen the paper in terms of methodology and clarity.
> > - **Regarding Q8:** I have to maintain my evaluation.
> >
> > In summary,  thank the authors for the detailed response and clarifications. I would increase my rating to positive.

---

### Official Review · Reviewer_rBWy · 2025-06-30

**Clarity:** 3
**Significance:** 2
**Originality:** 3
**Rating:** 5
**Confidence:** 3

**Summary:**

The paper introduces CogQA, a dataset for evaluating the use of specialized attention heads for cognitive functions similar to the brain in chain-of-thought reasoning tasks. The functions are created by decomposing problems from existing frameworks into sub-questions and classified into low-level and high-order functions. The authors use probes to predict sub-question class and identify which linear heads contribute to the classification.

**Questions:**

1. Have you investigated whether the question decomposition follows the implicit sequential reasoning assumption? Can it affect the quality of the generated dataset?

2. As the probe used is a 2-layers MLP, classification may come from non-linear interactions between several heads (across layers). Have you investigated this hypothesis?

3. Can you detail more precisely how you select the top-k most important tokens in the LLM's answer?

**Ethical Concerns:**

["NO or VERY MINOR ethics concerns only"]

**Final Justification:**

The discussion with the authors resolved my concerns and answered my questions and misconceptions about the paper.
 - The issues I had with the subquestion decomposition have been adressed
 - The concerns about the non-linear interactions that could be captured from probing have similarly been resolved

Overall, this is a very interesting work and I keep my positive score.

**Limitations:**

yes

**Quality:**

4

**Strengths And Weaknesses:**

The paper discusses an interesting problem and goes beyond what is done in most of the functional analysis literature by tackling CoT reasoning tasks instead of single-token prediction. The writing is clear and easy to follow and the experiments back up the claims made in the paper. While the findings are not fundamentally novel, I believe they are significant and of interest for the research community.

However, I have a few concerns:

1. The CoT decomposition is not generated by the evaluated model but by another model (GPT-4o when creating the dataset), one possible consequence is that these heads are not significantly used by the model in practice as it may not generate the same decomposition or may even not be able to plan the right reasoning steps.

2. Probing only tells that a head contains the necessary information for classification, and not that this head is actually used in the computation or that it executes the associated function. In particular, if my understanding is correct, the probe takes information from every layer at once (and not from a particular layer), so the classification may result from non-linear interactions between heads and even across layers (the quantity of noise in the head importance in Figure 2 supports this hypothesis).

3. The decomposition method divides a problem into a list of sub-questions, implicitly assuming that the reasoning structure is sequential while it could be graph-based (e.g. diamond shape: solve two independent subproblems and combine the results to solve the full task).

4. minor typo: final line of section 2.2 "for generating subqeustions"

---

> ### Author Rebuttal · Authors · 2025-07-31
>
> Dear Reviewer,
>
> Thank you for your constructive comments and suggestions. They have been invaluable in helping us enhance the quality and clarity of our paper. Please find our point-by-point responses to your concerns below.
>
> **Q1. The CoT decomposition is not generated by the evaluated model but by another model (GPT-4o when creating the dataset), one possible consequence is that these heads are not significantly used by the model in practice as it may not generate the same decomposition or may even not be able to plan the right reasoning steps.**
>
> **A.**  Our intention is to use chain-of-thought (CoT) not as an output of the model under analysis, but as a lens to study the model’s internal structure. The CoT decompositions are carefully designed and verified by human annotators to reflect the human-like cognitive steps required to solve the problem.
>
> Instead of prompting the model with the full question, we query it with individual subquestions, each aligned with a specific cognitive function. During prompting, previous subquestions and their ground-truth subanswers are provided as prior context. This setup allows us to observe which attention heads are activated as the model processes each cognitive step. In doing so, we can probe the internal mechanisms underlying distinct reasoning capabilities, offering insights into how different heads support specific cognitive functions—even if the model does not decompose problems this way under standard inference.
>
> **Q2. Probing only tells that a head contains the necessary information for classification, and not that this head is actually used in the computation or that it executes the associated function. In particular, if my understanding is correct, the probe takes information from every layer at once (and not from a particular layer), so the classification may result from non-linear interactions between heads and even across layers (the quantity of noise in the head importance in Figure 2 supports this hypothesis). As the probe used is a 2-layers MLP, classification may come from non-linear interactions between several heads (across layers). Have you investigated this hypothesis?**
>
> **A.** Thank you for raising this insightful point. Inspired by findings in neuroscience [1-2] where specific regions of the human brain are activated when performing certain tasks, we hypothesize that particular attention heads are similarly activated when a task mainly relies on a specific cognitive function. These heads exhibit distinct activation patterns that set them apart from other unrelated heads. shows more important for classification.
>
> Additionally, the interventions (positive and negative) complement probing by demonstrating the causal impact of those heads on task performance, suggesting that at least some of the probed signals are functionally utilized.
>
> In our initial experiments, we trained separate logistic regression classifiers for each head in each layer. However, results on Llama3-8B show that cognitive heads identified using only activation features and multiple independent classifiers did not outperform those selected by our current single-classifier approach in subsequent intervention experiments.
>
> [1] John R Anderson. Rules of the mind. Psychology Press, 2014.
>
> [2] Poldrack, Russell A. "Can cognitive processes be inferred from neuroimaging data?." Trends in cognitive sciences 10.2 (2006): 59-63.
>
> **Q3.The decomposition method divides a problem into a list of sub-questions, implicitly assuming that the reasoning structure is sequential while it could be graph-based (e.g. diamond shape: solve two independent subproblems and combine the results to solve the full task).**
>
> **A.** In constructing CogQA, annotators are instructed to ensure that each subquestion aligns with natural human reasoning. Specifically, if a subquestion depends on prior information—such as the question text or the answer—from another subquestion, the subquestion order must reflect this dependency. While some subquestions can be answered in parallel and are order-independent, others have prerequisite relationships that require a specific sequence.
>
> We acknowledge that the overall reasoning structure often forms a graph, where both sequential and parallel dependencies coexist.
> During LLM inference, we include the previous subquestions and their corresponding subanswers in the prompt as prior information. Thus, the critical factor is not the ordering alone, but whether the prompt provides the necessary context to answer the current subquestion accurately.
>
> **Q4. Can you detail more precisely how you select the top-k most important tokens in the LLM's answer?**
>
> **A.** We select the top-k most important tokens by prompting LLM.
> Given a reference answer and a predicted answer to a sub-question. We first predict whether the prediction answer is consistent to the reference answer, "True" and "False".
>
> If "True" is returned, extract and return the 5 most semantically important tokens from the prediction answer.
>    - These tokens should reflect the core meaning of the answer.
>    - Avoid common stopwords unless they are crucial to the semantics.
>
> **Q5. minor typo: final line of section 2.2 "for generating subqeustions"**
>
> **A.** Thank you for pointing out this typo. We will carefully proofread the manuscript to ensure clarity and correctness throughout.

---

> > ### Comment · Reviewer_rBWy · 2025-08-03
> >
> > I thank the authors for their response, it clarifies some of my concerns and misconceptions. I still have a few comments:
> >
> > Q1. I still think that probing a model's mechanisms following its own problem decomposition rather than a predefined one would be more interesting and impactful but I am happy with the authors' answer.
> >
> > Q2. Can you elaborate on what experiments you are referring to in this sentence and where they can be found in the paper?
> > > In our initial experiments, we trained separate logistic regression classifiers for each head in each layer. However, results on Llama3-8B show that cognitive heads identified using only activation features and multiple independent classifiers did not outperform those selected by our current single-classifier approach in subsequent intervention experiments.
> >
> > Q4. Can you further detail how you assess if a token is semantically relevant in the prediction? Are you using attention scores?

---

> ### Author Response · Authors · 2025-08-04
> **Response to Reviewer rBWy**
>
> Dear Reviewer,
>
> We sincerely thank the reviewer for the thoughtful comments and are glad that our response helped clarify some concerns. Below we address the remaining points in more detail:
>
> **Q1. Can you elaborate on what experiments you are referring to in this sentence and where they can be found in the paper?**
>
> **A.** We apologize for the confusion. Our initial experiments were conducted on Llama3-8B-Instruct, but the final version of the paper uses the newer Llama3.1-8B-Instruct. As a result, those earlier experimental results are not included in the Appendix.
>
> In the revised setting, we performed comparative experiments using two classification methods—MLP and Logistic Regression—on Llama3.1-8B-Instruct. We evaluated their effectiveness from two perspectives: 1) heatmap visualization and 2) negative intervention experiments (masked the top-30 cognitive heads identified by different methods). From the heatmaps, we observed that cognitive head activations exhibit more noise than those shown in Figure 2 (omitted due to space constraints).
> The results on Llama3.1-8B-Instruct (shown in the following table) demonstrate that masking the heads identified by our method leads to a greater performance drop, indicating the effectiveness of our method.
>
> **Table: The performance of LLM by masking related cognitive heads from different classification methods.**
>
> | Classifier | Retrieval (comet) | Retrieval (acc) | Math (comet) | Math (acc) |
> |-|-|-|-|-|
> | Logistic Regression | 88.69  | 81.68 | 93.61  | 73.63  |
> | MLP (ours)    | 70.05  | 46.47  | 89.32 | 67.16  |
>
>
> **Q2. Can you further detail how you assess if a token is semantically relevant in the prediction? Are you using attention scores?**
>
> **A.** We prompt GPT-o4-mini to select the five most semantically important tokens in the predicted answer. This selection is performed internally by the model using its attention mechanism, which implicitly captures token-level relevance. Given GPT-o4-mini's strong reasoning capability, this serves as a reliable proxy for identifying semantically meaningful tokens.
>
> The following table presents the results of masking 30 and 50 retrieval heads, based on classifications using activation features extracted with different token selection strategies: first is the first token, last is the last token, meaning\_first is the first meaning token (excluding formatting), top-k is the top-k most semantically important tokens, full is all tokens in the answer. We observe that using top-k token leads to the most significant performance drop when masking the top-30 identified heads, indicating higher precision in identifying retrieval-relevant heads. Interestingly, last, meaning\_first, full, and top-k exhibit similar performance trends when masking 50 heads. This is because different tokens in the output contribute to answering the question, and as the number of masked cognitive heads increases, the influence of token using decreases. Additionally, for Retrieval, the full answer is usually meaningful, whereas others like Math Calculation require semantically meaningful tokens. Based on these results, we choose top-k as our final setting. Notably, the results in the Appendix report only the case of masking the top-50 heads. We will update the Appendix with the full comparison in the final version.
>
> **Table: The performance of LLM by masking related cognitive heads identified with different tokens.**
>
> | Model         | Head\_num | Token\_use     | Retrieval (comet) | Retrieval (acc) | Math (comet) | Math (acc) |
> |-|-|-|-|-|-|-|
> | Llama3.1-8B   | 30  | first   | 90.51   | 83.53    | 91.13   | 73.13   |
> | Llama3.1-8B   | 30  | last   | 86.86     | 81.76    | 90.04   | 68.66    |
> | Llama3.1-8B   | 30  | meaning_first  | 88.13   | 79.41   | 89.72        | 68.66    |
> | Llama3.1-8B   | 30  | full  | 73.93     | 47.06            | 89.92        | 69.15       |
> | Llama3.1-8B   | 30  | top-k  | **70.05**          | **46.47**        | **89.32**        | **67.16**   |
> | Llama3.1-8B   | 50 | first  | 93.28    | 89.41   | 94.46  | 89.57    |
> | Llama3.1-8B   | 50 | last  | 64.39    | 41.18 | 92.05  | 70.15       |
> | Llama3.1-8B   | 50 | meaning_first  | 62.90    | 34.12    | 84.60        | 60.69       |
> | Llama3.1-8B   | 50 | full  | 46.20       | 11.76      | 89.01      | 78.11       |
> | Llama3.1-8B   | 50  | top-k | 65.64    | 47.76     | 89.65        | 70.15       |
>
> We thank the reviewer for the insightful comment: "I still think that probing a model's mechanisms following its own problem decomposition rather than a predefined one would be more interesting and impactful, but I am happy with the authors' answer." We agree that this is an interesting and important direction for future research. In particular, integrating this perspective with our framework may help explore questions such as whether a model possesses distinct cognitive abilities (heads) that it does not actively utilize.
>
> Thanks for your time.
>
> Best Regards,
>
> The Authors

---

> > ### Author Response · Authors · 2025-08-05
> > **Response to Reviewer rBWy**
> >
> > Dear Reviewer,
> >
> > For top-k tokens, we give examples for different cognitive functions for Llama3.1-8B-instruct as below:
> >
> > *main question: "Given the sentence \"A surfboarder catches the waves.\" can we conclude that \"A surfboarder in the water.\"?\\nOptions:\\n- yes\\n- it is not possible to tell\\n- no"*
> >
> > subquestion: "What is typically required for a surfboarder to catch waves?"
> >
> > cognitive function: "Knowledge Recall"
> >
> > answer: "The surfboarder needs to be in the water."
> >
> > selected tokens: ['surfboarder', 'needs', 'be', 'in', 'water']
> >
> > *main question: "Is the following a factual statement?\\n\"Due to its high density, countries around the world use Palladium to mint coins.\"\\nOptions:\\n- yes\\n- no"*
> >
> > subquestion: "What is the statement in question?"
> >
> > answer: "The statement in question is: Due to its high density, countries around the world use Palladium to mint coins."
> >
> > cognitive function: "Retrieval"
> >
> > selected tokens: ['high', 'density', 'Palladium', 'mint', 'coins']
> >
> > *main question: "A one-year subscription to a newspaper is offered with a 45\% discount. How much does the discounted subscription cost if a subscription normally costs \$80?"*
> >
> > subquestion: "How much is the discount amount in dollars for the subscription?"
> >
> > answer: "36"
> >
> > cognitive function: "Math Calculation"
> >
> > selected tokens: ['36']
> >
> > We can see that the selected tokens semantically represent the answer. Note that we use all tokens when the number of tokens is fewer than 5.
> >
> > Thank you once again for your thoughtful review and consideration.
> >
> > Best regards,
> >
> > The Authors

---

> > > ### Comment · Reviewer_rBWy · 2025-08-05
> > >
> > > I thank the authors for their extensive response. My follow-up questions have been adequately answered so I maintain my positive assessment of the paper.

---

### Decision · Program_Chairs · 2025-09-17

**Decision:**

Accept (poster)

**Comment:**

This work mechanistically analyzes roles and behaviors of attention heads inside LLMs and introduces a dataset to divide complex questions into multiple steps with retrieval or logical reasoning cognitive functions. Furthermore, they locate universal sparse, interactive, and hierarchical cognitive attention heads across different LLMs. In addition, removing or augmenting those heads can affect reasoning tasks.

The main strength of this work is to conduct a comprehensive and mechanistic analysis of cognitive attention heads on their proposed dataset across different LLMs. In addition, the following located attention head intervention experiment demonstrates how those heads affect reasoning tasks.

For weakness, several reviewers raised concerns about the cognitive question decomposition setting, the vague definition of distinct cognitive functions, and the analysis of how attention heads behave (layered functional organization) by masking cognitive function relevant or random masking leading to different performance drops.

In general, different reviewers’ questions were well solved regarding those concerns above during the discussion.

Based on reviewer-author discussions, I recommend that authors add those extra explanations into the camera-ready version, such as how LLMs process the CoT decomposition evaluated by another model, what if reasoning structure is not sequential, new experimental results for probing, how to evaluate and select semantically relevant tokens, etc.

To summarize, I would recommend this work as poster for NeurIPS.